# Analysis of the Dynamic Thermal Barrier in Building Envelopes

Veronika Mučková [1], Daniel Kalús [1], Daniela Koudelková [1], Mária Kurčová [1], Zuzana Straková [1], Martin Sokol [1,*], Rastislav Ingeli [2] and Patrik Šťastný [3]

[1] Department of Building Services, Faculty of Civil Engineering, Slovak University of Technology, Radlinského 11, 81005 Bratislava, Slovakia
[2] Department of Building Construction, Faculty of Civil Engineering, Slovak University of Technology, Radlinského 11, 81005 Bratislava, Slovakia
[3] Department of Building Technology, Faculty of Civil Engineering, Slovak University of Technology Radlinského 11, 81005 Bratislava, Slovakia
[*] Correspondence: martin.sokol@stuba.sk; Tel.: +421-903-220-939

**Abstract:** This article focuses on the investigation of the dynamic thermal barrier (TB) and dynamic thermal resistance (DTR) of the building envelope. The aim is to analyze the DTR as a function of the temperature change of the heat transfer medium supplied to the dynamic TB layer and to determine the energy potential of several materially different fragments of the building envelope. The functions of TB and DTR depend on the uniform and continuous maintenance of temperature in a given layer of the building structure. The methodology is based on the analysis and synthesis of thermal resistance calculation, wall heating, and computer simulation. The research results show that the relatively low mean temperature of the heat transfer medium of approximately $\theta_m$ = 17 °C delivered to the TB layer represents $R_{DTR}$ = up to 30 ((m²·K)/W) for an equivalent dynamic thermal insulation thickness of 1000 mm for a required standard resistance of $R_{STANDARD}$ = 6.50 ((m²·K)/W) of the individual fragments analyzed with static thermal insulation of 65 to 210 mm. The energy potential of a thermal barrier (TB) represents an increase of approximately 500% in the thermal resistance and up to 1500% in the thickness of the dynamic thermal insulation. Further research on the dynamic thermal barrier and verification of the results of the parametric study will continue with comprehensive computer simulations and experimental measurements on the test cell.

**Keywords:** active thermal protection; thermal barrier; dynamic thermal resistance; temperature progression; parametric study; thermal simulation

## 1. Introduction

As defined by Kalús [1], "*Active thermal protection (ATP) is a dynamic process characteristic of building structures with integrated energy-active elements that are characterized by one or more functions in different operating modes of energy systems and heat sources: thermal barrier (TB), large-scale heating/cooling, heat/cooling storage, solar and ambient energy capture, heat/cooling recovery, and other combinations.*"

The principle of the ATP is to install pipes of the heat exchange surface between the load-bearing and heat-insulating part and/or on the exterior side of the building envelope. The heat transfer medium is usually temperature-treated water or another liquid, but it can also be air [2]. Dry and wet methods of integrating the ATP system are known. The wet method is more difficult to implement, and in this method, pipes of the heat exchange surface are placed in the "wet part" of the load-bearing part of the building envelope (reinforced concrete wall/roof) or mortar bed (masonry wall) [3]. The dry method of making ATP is faster and simpler and consists of placing the pipes of the heat exchange surface between the load-bearing and heat-insulating part of the building envelope or into the thermal insulation as a part of the thermal insulation panels of the contact insulation system [4].

Depending on the ATP function and the composition of the building envelope, the mean temperature of the heat transfer medium can range from as low as 6 °C for the TB function (timber house with thermal insulation envelope panels) with a high dynamic thermal resistance $R_{DTR}$ = 10.487 ((m²·K)/W), corresponding to an external static thermal insulation of approximately 200 mm, up to between 25 and 30 °C for the large-scale radiant heating function (reinforced concrete thermal insulated building envelope). This contribution focuses on the research area of building wall structures with integrated energy active elements and dynamic thermal resistance, specifically ATP in the TB function. The principle of dynamic thermal resistance (DTR) of the building envelope is the controlled delivery of heat/cool to the ATP heat transfer layer over time, which adjusts the heat/cool transfer through the building structure as required while limiting the heat loss/gain of the building. Dynamic thermal insulation, therefore, can use a heat transfer agent to adjust the desired temperature in the ATP layer over time and, according to the DTR size requirement, thereby eliminating the thickness of static thermal insulation.

The aim of the research described in this contribution is a parametric study of the dynamic thermal resistance, simulation of the progression of the temperature in the individual layers of the structure, and determination of the energy potential of several materially different fragments of building envelope wall structures with integrated energy-active elements under different input data of physical variables.

## 2. Current Status of Technical Solutions and Overview of Research Work in the Field of Active Thermal Protection

Active thermal protection in various functions (TB, wall heating/cooling, heat/cool accumulation, etc.) is being investigated by several researchers worldwide. In the following section, we present the current state of the technical solutions and an overview of important studies in this area of research, which our research builds on.

### 2.1. Inspirational Technical Solution of Our Research

Among the first scientists, developers, and researchers in the field of active thermal protection and thermal barrier is Dipl.-Ing., Phys. Edmond D. Krecké from Luxembourg, who combined the advantages of building constructions with integrated energy-active elements with RES-based energy systems and commercially called this system ®Isomax/®Terrasol Technologien. Since about the 1990s, he has constructed several buildings around the world using this technology with thermal barriers, Figures 1–4, [5–7]. This combined building energy system inspired us to research building structures with integrated energy-active elements.

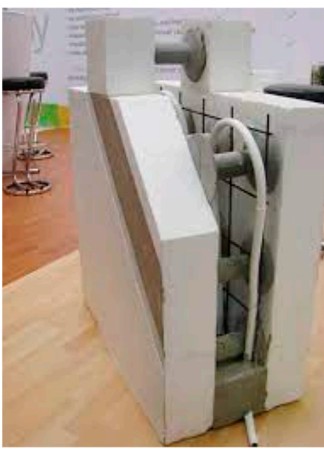

**Figure 1.** Lost formwork for reinforced concrete building envelope ISOMAX [5].

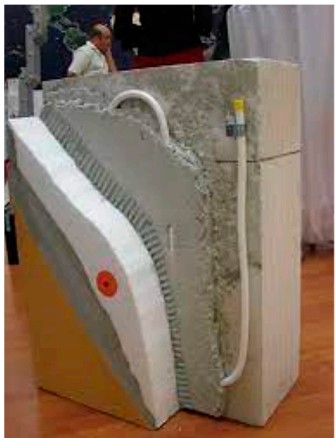

**Figure 2.** Application of TB on porous concrete block wall [5].

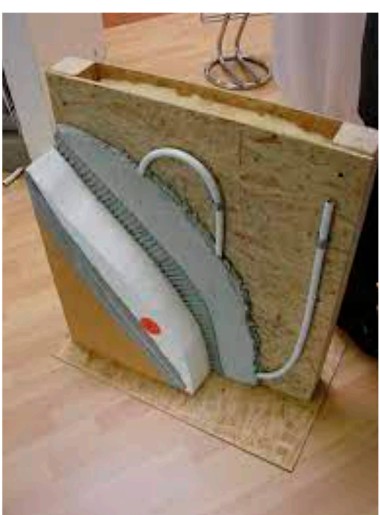

**Figure 3.** Application of TB on the wall of a timber building [5].

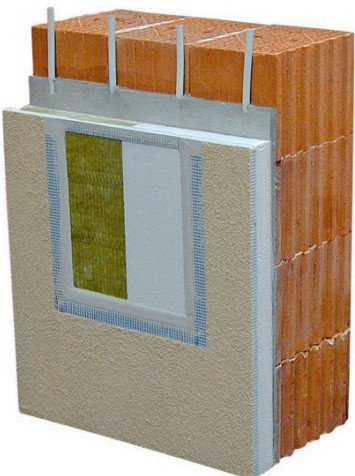

**Figure 4.** Application of TB on a brick wall [5].

*2.2. Scientific and Professional Work in the Field of ATP*

Many researchers from all over the world are working in the field of active thermal protection research. In this section, we present important outputs and results published in scientific articles and studies.

Koenders, S. J. M., Loonen, R. C. G. M., and Hensen, J. L. M. [8], 2018, investigated the performance of a new type of closed dynamic forced convection insulation system on a simulation model developed in EnergyPlus. The first results showed that a nine times lower value of the heat transfer coefficient U can be achieved compared to a conventional insulation system. The best results in terms of energy consumption and indoor thermal comfort were demonstrated when the dynamic insulation was used on a façade structure with a heavy inner partition and a light outer partition.

Krzaczek, M., Florczuk, J., and Tejchman, J. [9], 2019, report comprehensive results of measurements in a two-story family house in northern Poland over 17 months. Control software using fuzzy logic was designed to optimize the operation of the energy system. The measurements confirmed the high efficiency of the thermal barrier and its ability to efficiently use low-energy heat and cold sources to ensure thermal comfort.

Kisilewicz, T., Fedorczak-Cisak, M., and Barkanyi, T. [10], 2019, investigated the possibility to what extent an active thermal protection system can replace commonly used passive thermal insulation systems. During the course of three years, the study was conducted in an experimental residential structure in Nyiregyháza, Hungary, that had a ground heat exchanger that was directly connected to a wall heat exchanger.

Fawaier, M. and Bokor, B. [11], 2022, aimed to unify the research carried out in the field of dynamic building insulation systems as follows: by categorizing the literature according to the type of structures used, the approach to the research problem, and the parameters investigated. Based on the historical development of ATP methods across time, the authors examined several mathematical models, experimental research, and numerical simulations. They also investigated innovative ATP designs that utilize air.

Shen, J., Wang, Z., Luo, Y., Jiang, X., Zhao, H., and Tian, Z. [12], 2022, investigated an active building envelope system designed to redistribute the heat gained from the absorption of solar heat gains by the south façade to the exterior walls of the north façade based on a mathematical model. The thermal performance of the system was simulated for five climatic regions of China and then compared with a conventional wall system.

Yan, B., Han, X., Malkawi, A., Dokka, T. H., Howard, P., Knowles, J., and Edwards, K. [13], 2022, comprehensively evaluated the operation of a thermally active building system (TABS) with large thermal inertia combined with natural ventilation as a dynamic system. Measurements were conducted in an experimental office building in Massachusetts during the COVID-19 pandemic when the building was unoccupied. They demonstrated the high effectiveness of the system in managing thermal comfort as well as its high energy efficiency.

Junasová, B., Krajčík, M., Šikula, O., Arıcı, M., and Šimko, M. [14], 2022, investigated various design modifications of the radiant ceiling and wall systems for heating and cooling suitable for both new and retrofitted buildings. A validated mathematical model was used to calculate the heat transfer. The highest increase in power per 1 cm was for a pipe pitch of 60 mm. The small spacing increased the system's performance per energy delivered by creating a uniform surface temperature while reducing response time. The highest performance of all the cases studied was achieved by attaching a metal slat to a pipe embedded in insulation.

Šimko, M., Petráš, D., Krajčík, M., and Szabó, D. [15], 2022, investigated the thermal performance, surface temperatures, and heat transfer temperatures of a wall cooling system with a pipe connected to a part of the wall made of thermal insulation blocks. The measurements were carried out in a climate chamber. Utilizing 2D numerical simulations, the sensitivity of the thermal performance to various design parameters was examined. Due to the thermal resistance of the thermal insulation, the air temperature had a minimal impact on the cooling effectiveness. The performance of the cooling was not significantly impacted by the heat transfer coefficient between the water and the wall or between the wall and the outside. On the other hand, an increase in room temperature of only 1 °C raised the thermal performance by approximately 7 W/m², while an increase in the heat

transfer coefficient between the wall and the interior by 1 W/(m².K) raised the performance by 3 to 4 W/m².

Chen S., Yang Y., and Chang T. [16], 2023, examined the hydraulic thermal barrier (HTB), which enables the building envelope to be gradually viewed as a multifunctional element, as a way to change the characteristics of thermal insulation solutions from ones that are high in carbon to ones that are low in carbon. Yet, poor operational management, design, and construction can result in issues such as high costs and low efficiency, as well as the exact opposite technical impacts. A thorough uncertainty analysis is numerically carried out in this article. The findings of the uncertainty analysis demonstrated that, when the variables are properly chosen, the proper usage of HTB could greatly reduce the heat gains that a conventional air-conditioning system must contend with and could even have the technical effect of supplementary cooling. To assist in generating a continuous thermal buffer zone in the building envelope, tube spacing should ideally be set between 100 and 250 mm.

### 2.3. Scientific Works in the Field of ATP Computer Simulations

For mathematical modeling and computer simulations, we were inspired by the following scientific works.

Q. Zhu, X. Xu, J. Wang, and F. Xiao [17], 2014, presented a dynamic simplified thermal model of a structure with active thermal protection of a building with embedded ductwork and identified the parameters of the simplified model based on the analysis of the frequency characteristics. Various case studies are presented to verify the accuracy of the simplified models and the effectiveness of model parameter identification. An optimal simplified model can be developed and provide a reasonable and accurate prediction of the performance independent of the composition of the active structure. However, the accuracy of the model may vary depending on the physical properties of the wall.

Wu XZ, Zhao JN, Olesen BW, Fang L, and Wang FH. [18], 2015, proposed and developed a new simplified model to calculate the surface temperature and heat transfer of radiant floor heating and cooling using the conduction shape factor. The proposed model was verified using measured data from references. The findings indicated that, for the radiant floor heating system, for an average water temperature between 40 °C and 60 °C, the greatest disparities between the calculated surface temperature and heat transfer using the suggested model and the observed data were 0.8 °C and 8.1 W/m², respectively. In this work, the suggested model was also validated using numerically generated data.

J. Xie, X. Xu, A. Li, and Q. Zhu [19], 2015, presented an experimental verification of the finite-difference frequency-domain (FDFD) model of a building with an active shell with integrated ductwork in the time domain using the Fourier series analysis. The thermal response of the active building envelope at predetermined boundary conditions has been measured using a test rig. For testing, various supply water temperatures were applied. The findings demonstrate that the time-domain thermal responses of the active building envelope predicted by the FDFD model and experimental observations are in good agreement. The outcomes also demonstrate how simple and effective it is to use Fourier series analysis to validate the FDFD model over time.

Q. Zhu, A. Li, J. Xie, W. Li, and X. Xu [20], 2016, dealt with the experimental verification of a semi-dynamic simplified model of an active building envelope with embedded ductwork. The simulation results were compared with measurements in real conditions and confirmed that the semi-dynamic simplified model could predict the semi-steady or dynamic thermal properties of the active building envelope very well.

Lydon, G. P., Caranovic, S., Hischier, I., and Schlueter, A. [21], 2019, discussed a coupled simulation for the design of a heating and cooling system embedded in a lightweight roof structure. A parametric geometric model has been developed for the application of the piping system to a complex roof shape. The study focuses on the modeling techniques used in the energy sector to facilitate the creation of a digital twin for the building's multifunctional component. Building physics concerns are dealt with, and initial system

performance is ensured using high-resolution analysis. The lower resolution method is used to add system characteristics to an industrial simulation model of the entire building using the input data from the first two processes. The creation of the first control techniques for the new multifunctional element is possible in this last step.

## 3. Methodology

The research described in this paper aims to analyze the dynamic thermal barrier in building envelopes, which is characterized by the dynamic thermal resistance dependent on the temperature change of the heat transfer medium supplied to the dynamic TB layer, and to determine the energy potential of several materially different fragments of the building envelope. The TB and DTR functions depend on a uniform and continuous temperature maintenance in a given layer of the building structure. The research methodology is based on the analysis and synthesis of procedures for calculating thermal resistance and heat transfer coefficient according to EN 73 0540, wall heating according to EN 1264-(1-5), and computer simulation in ANSYS.

### 3.1. Calculation Procedure for Thermal Resistance R and Heat Transfer Coefficient U according to EN 73 0540

Applying a thermal barrier to the structure eliminates the thickness of thermal insulation on the exterior side of the envelope. This means that we can reduce heat losses or gains. To be able to reduce the above-mentioned parameters, we need to know the temperature behavior in the structure, the most important being the temperature between the thermal insulation and the load-bearing layer of the structure. Thermal barrier pipes are applied between these layers. To be able to calculate the individual temperatures between the layers and to plot the temperature waveform in a fragment of a multilayer structure, we need to know the heat transfer coefficient U (W/(m²·K)) and the thermal resistance of the building structure R ((m²·K)/W).

The thermal resistance of the j-th structure is calculated using the formula [22,23]:

$$R_i = \frac{d_j}{\lambda_j} \tag{1}$$

where $R_i$ is the thermal resistance of the j-th layer of the structure ((m²·K)/W), $d_j$ thickness of the j-th layer of the structure (m), and $\lambda_j$ coefficient of thermal conductivity of the j-th layer of the structure (W/(m·K)).

The thermal resistance of a multilayer structure $R_c$ ((m²·K)/W) is calculated using the formula [22,23]:

$$R_c = \sum R_i \tag{2}$$

$$R = R_{si} + R_c + R_{se} \tag{3}$$

where R thermal resistance of the structure ((m²·K)/W), $R_c$ total thermal resistance of the structure ((m²·K)/W), $R_j$ is the thermal resistance of the j-th layer of the structure ((m²·K)/W), $R_{si}$ thermal resistance to heat transfer at the internal surface of the structure ((m²·K)/W), and $R_{se}$ thermal resistance to heat transfer at the external surface of the structure ((m²·K)/W).

The heat transfer coefficient of a multilayer structure U (W/(m²·K)) is calculated using the formula [22,23]:

$$U = \frac{1}{R} = 1/(R_{si} + R_c + R_{se}) \tag{4}$$

where U is the heat transfer coefficient of the structure (W/(m²·K)/), R thermal resistance of the structure ((m²·K)/W), $R_{si}$ thermal resistance to heat transfer at the internal surface of the structure ((m²·K)/W), and $R_{se}$ thermal resistance to heat transfer at the external surface of the structure ((m²·K)/W).

The temperature in the j-th layer of the structure $\theta_j$ (°C) is calculated using the formula [22,23]:

$$\theta_j = \theta_i - U \times (\theta_i - \theta_e) \times (R_{si} + \textstyle\sum R_j) \tag{5}$$

where U heat transfer coefficient of the structure (W/(m²·K)), $R_{si}$ thermal resistance to heat transfer at the internal surface of the structure ((m²·K)/W), $\sum R_j$ sum of thermal resistances of the j-th layers of the structure ((m²·K)/W), $\theta_e$ outdoor design temperature in winter (°C), $\theta_i$ internal design temperature (°C), and $\theta_j$ is the temperature in the j-th layer of the structure (°C).

In our research, we assume the wall component is a 3D system, that is, the heat transfer equation for the transient conditions in the Cartesian coordinate system is written as [24]:

$$C_p \rho \frac{\partial T}{\partial t} = \lambda \left( \frac{\partial^2 T}{\partial x} + \frac{\partial^2 T}{\partial y} + \frac{\partial^2 T}{\partial z} \right) \tag{6}$$

where $C_p$ is the specific heat under constant pressure (kJ/(kg·K)), T the temperature (K), t the time (s), and $\rho$ the density of the wall layer material (kJ/m³)).

To use or calculate the heat transfer equation, we need to specify the boundary conditions. In our case, the perimeter wall structure separates the interior space temperature $T_i$ (K) from the ambient conditions. This means that the boundary conditions of the $S_i$ (m²) and $S_e$ (m²) surfaces are defined by Newton's law. Consequently, the heat transfer coefficient by radiation and convection, respectively, is defined by the rate of heat exchange by convection and radiation on the interior surface $S_i$ (m²). That is, the boundary conditions on the $S_i$ (m²) surface are defined using the formula [24]:

$$\left. \frac{\partial T(t)}{\partial x} \right|_{s_i} = h_i \left[ T_{Fi}(t) - T_i \right] \tag{7}$$

where $h_i$ the convective/radiative heat transfer coefficient on the internal surface (W/(m²·K)), $T_i$ the internal air temperature (K), $T_{Fi}(t)$ the internal surface temperature (K), and $\lambda$ is the thermal conductivity (W/(m·K)).

In a structure, there is a heat exchange between the external $S_e$ (m²) surface and the external environment. The heat exchange is composed of convection and radiation, and these two components must be considered separately. Radiation is defined by the solar air temperature, and convection is defined by the convective heat transfer coefficient. We would define the solar air temperature $T_i$ (K) as the fictitious temperature of the outdoor air that, in the absence of radiative exchange at the exterior surface of the roof or wall, would provide the same rate of heat transfer through the wall or roof as the actual combined heat transfer mechanism between the sun, the surface of the roof or wall, and the outdoor air and environment. Since the ambient conditions are variable, we can define the boundary conditions on the external surface $S_e$ (m²) using the formula [24]:

$$\left. \lambda \frac{\partial T}{\partial x} \right|_{s_e} = h_e\,(t)[T_e(t) - T_{Fe}(t)] \tag{8}$$

where $h_e(t)$ is the convective heat transfer coefficient on the external surface (W/(m²·K)), $T_{Fe}(t)$ is the external surface temperature (K), and $T_e(t)$ is the sol-air temperature (K).

The boundary conditions on the adiabatic surfaces $S_{a1}$ (m²) and $S_{a2}$ (m²) are defined using the formula [24]:

$$q(t)\,|\,S_{a1} = 0 \tag{9}$$

and

$$q(t)\,|\,S_{a2} = 0 \tag{10}$$

where q(t) is the heat flux normal to the surface (W/m²).

### 3.2. Calculation Procedure for Wall Heating according to EN 1264-(1-5)

To calculate the thermal performance of active thermal protection, the boundary conditions (location, outdoor temperature, indoor temperature, etc.) are required. The next important step is to calculate the heat loss and heat input. The wall heating performance is designed for individual room heat input according to STN EN 1264-(1-5), [25–29].

To determine the specific heat capacity q (W/m²) of the wall surface, the following parameters are required [25–29]:

- Heating pipe spacing;
- The thickness $s_u$ and the thermal conductivity $\lambda_E$ of the wall layer in front of the heating tubes towards the interior;
- The thermal resistance of the surface covering $R_{\lambda,B}$ of the wall;
- The outer diameter of the heating tubes $D = d_a$, possibly with coating $D = d_M$, and the thermal conductivity of the heating tubes $\lambda_R$ or coating $\lambda_M$;
- The contact between the tubes and the heat pipe elements or spreading layer is characterized by the coefficient $a_K$.

The specific heat capacity is calculated according to the equation [25–29]:

$$q = B \times \prod_i (a_i^{m_i}) \times \Delta\theta_H \tag{11}$$

where q is the specific heat capacity (W/m²), B is the system dependent coefficient (W/(m²·K)), and $\prod_i(a_i^{m_i})$ is the power product combining the design parameters between each other. The design parameters are calculated as follows [25–29]:

$$\Delta\theta_H = \frac{\theta_V - \theta_R}{\ln\frac{\theta_V - \theta_i}{\theta_R - \theta_i}} \tag{12}$$

where $\Delta\theta_H$ is the average temperature of the heating medium (°C), $\theta_V$ is the supply temperature of the heating medium (°C), $\theta_R$ is the return temperature of the heating medium (°C), and $\theta_i$ is the nominal indoor temperature of the room (°C).

The specific heat capacity is proportional to $(\Delta\theta_H)_n$; exponent n has values according to theoretical results confirmed by experiments: $1.0 < n < 1.01$. Within the limits of sufficient precision, a value is used, $n = 1$, [25–29].

In STN EN 1264, there are different types of compositions of structures. For the type of structure we are analyzing, the most relevant standard structure type is B, Figure 5.

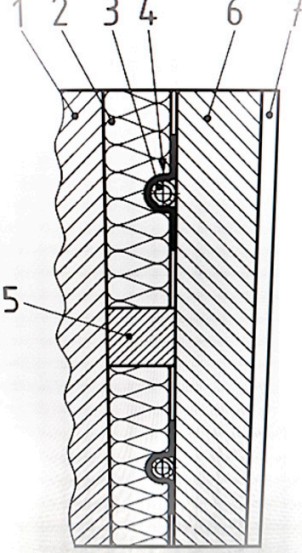

**Figure 5.** Wall system with pipes in the insulation layer with thermal diffusion devices, type B (STN EN 1264-1). 1—wall structure, 2—insulation layer, 3—piping, 4—thermal diffusion device, 5—fixing supports, 6—thermal diffusion layer, and 7—surface covering.

For a type B system, the specific heat capacity q (W/m²) is calculated according to the relation [25–29]:

$$q = B \times a_B \times a_T^{m_T} \times a_u^{m_u} \times a_{WL} \times a_k \times \Delta\theta_H \tag{13}$$

where q is the specific heat capacity (W/m²) and B is the system-dependent coefficient (W/(m²·K)). The application for this system is as follows [25–29]:

$$a_B = \frac{1}{1 + B \times a_u \times a_T^{m_T} \times a_{WL} \times a_K \times R_{\lambda,B} \times f(L)} \tag{14}$$

Herewith [25–29],

$$F(T) = 1 + 0.44\ \sqrt{L}., \tag{15}$$

where $a_B$ is the covering coefficient (−); $a_T$ is the tube spacing coefficient (−), Table 1; $a_T = f(s_u/\lambda_E)$; $\lambda_E$ is the thermal conductivity of the spreading layer (W/(m·K)); $s_u$ is the thickness of the spreading layer over the tubes (m); and $m_i$ is the exponents to calculate the characteristic curves ($m_T$, $m_u$) (−). This holds as follows [25–29]:

$$m_T = 1 - \frac{L}{0.075}\quad \text{suits when } 0.050\ m \le L \le 0.375\ m \tag{16}$$

$$m_u = 100\ (0.045 - s_u) \text{ suits when } s_u \ge 0.010\ m \tag{17}$$

where L is the spacing of the heating tubes (m) and $s_u$ is the thickness of the spreading layer above the tubes (m).

**Table 1.** Pipe spacing coefficient $a_T$ versus pipe spacing $L$ for type B systems, [25–29].

| $s_u/\lambda_E$ (m²·K)/W | 0.01 | 0.02 | 0.03 | 0.04 | 0.05 | 0.06 | 0.08 | 0.10 | 0.15 | 0.18 |
|---|---|---|---|---|---|---|---|---|---|---|
| $a_T$ | 1.103 | 1.100 | 1.097 | 1.093 | 1.091 | 1.088 | 1.082 | 1.075 | 1.064 | 1.059 |

If the pipe spacing is L > 0.375 m, the specific power is calculated from the formula [25-29]:

$$q = q_{0.375} \times \frac{0.375}{L} \tag{18}$$

where q is the specific heat capacity (W/m²) and $q_{0.375}$ is the specific heat capacity (W/m²) calculated at pipe spacing L = 0.375 m. The covering factor is calculated as follows [25–29]:

$$a_u = \frac{\frac{1}{\alpha} + \frac{s_{u,0}}{\lambda_{u,0}}}{\frac{1}{\alpha} + \frac{s_u}{\lambda_E}} \tag{19}$$

where $a_u$ is the covering factor (-); $\alpha$ = 10.8 W/(m²·K); $\lambda_{u,0}$ = 1 W/(m²·K); $s_{u,0}$ = 0.045 m; $a_k$ is the correction coefficient of coupling in compliance, Table 2; $a_k = f(T)$; $a_{WL}$ is the thermal conductivity coefficient; Table 3, $a_{WL} = f(K_{WL}, L, D)$; and $\Delta\theta_H$ is the average temperature of the heating substance (°C), Formula (12) [25–29].

**Table 2.** Contact correction factor $a_k$ as a function of pipe spacing $L$ for type B systems [25–29].

| $L$ (m) | 0.05 | 0.075 | 0.1 | 0.15 | 0.20 | 0.225 | 0.30 | 0.375 | 0.45 |
|---|---|---|---|---|---|---|---|---|---|
| $a_K$ | 1.00 | 0.99 | 0.98 | 0.95 | 0.92 | 0.90 | 0.82 | 0.72 | 0.60 |

**Table 3.** Thermal conductivity coefficient $a_{WL}$ versus pipe spacing $L$, outer diameter $D$, and characteristic $K_{WL}$ value for type B systems ($K_{WL} = 0$) [25–29].

| $D$ (m) | 0.022 | 0.020 | 0.018 | 0.016 | 0.014 |
|---|---|---|---|---|---|
| $L$ (m) | | | $a_{WL}$ | | |
| 0.05 | 0.96 | 0.93 | 0.90 | 0.86 | 0.82 |
| 0.075 | 0.80 | 0.754 | 0.70 | 0.644 | 0.59 |
| 0.10 | 0.658 | 0.617 | 0.576 | 0.533 | 0.488 |
| 0.15 | 0.505 | 0.47 | 0.444 | 0.415 | 0.387 |
| 0.20 | 0.422 | 0.40 | 0.379 | 0.357 | 0.337 |
| 0.225 | 0.396 | 0.376 | 0.357 | 0.34 | 0.32 |
| 0.30 | 0.344 | 0.33 | 0.315 | 0.30 | 0.288 |
| 0.375 | 0.312 | 0.30 | 0.29 | 0.278 | 0.266 |
| 0.450 | 0.30 | 0.29 | 0.28 | 0.264 | 0.25 |

Tables for $K_{WL}$ = 0.1, 0.2, 0.3, 0.4, 0.5, and above are seen STN EN 1264-2, where $K_{WL}$ is the coefficient of the heat conducting element for type B (-) systems and is expressed as follows [25–29]:

$$K_{WL} = \frac{s_{WL} \times \lambda_{WL} + b_u \times s_u \times \lambda_E}{0.125}$$

(20)

where $b_u = f(L)$—Table 4, $s_{WL} \times \lambda_{WL}$ = product of thickness and thermal conductivity of the heat conducting element, and $s_u \times \lambda_E$ = product of thickness and thermal conductivity of the spreading layer.

**Table 4.** Coefficient $b_u$ versus pipe spacing $L$ for type B systems [25–29].

| $L$ (m) | 0.05 | 0.075 | 0.1 | 0.15 | 0.20 | 0.225 | 0.30 | 0.375 | 0.45 |
|---|---|---|---|---|---|---|---|---|---|
| $b_u$ | 1.00 | 1.00 | 1.00 | 0.70 | 0.50 | 0.43 | 0.25 | 0.10 | 0.00 |

### 3.3. Computer Simulation Procedure—ANSYS

We describe the computer simulation procedure by creating a 2D model of a fragment of the perimeter wall structure of a prefabricated timber building with integrated energy-active elements as a function of the thermal barrier depending on various input variable factors.

The first step is to create a mathematical-physical model, where the 2D model of the fragment to be analyzed is shown, and to process the mathematical-physical properties of the different layers of the structure into a clear table, as shown in Figure 6. The simulation procedure is shown in Figure 7. The first step of the simulation was to specify the material characteristics of the individual layers of the structure, make a model of the fragment in ANSYS, and generate a mesh with a suitable element size (in our case, $5^{-003}$ m), as seen in Figure 8. Next, it was necessary to determine the boundary conditions, where the first condition was the convection condition for the interior, the second was the convection condition for the exterior, and the last condition was the water temperature in the pipes of 6 °C.

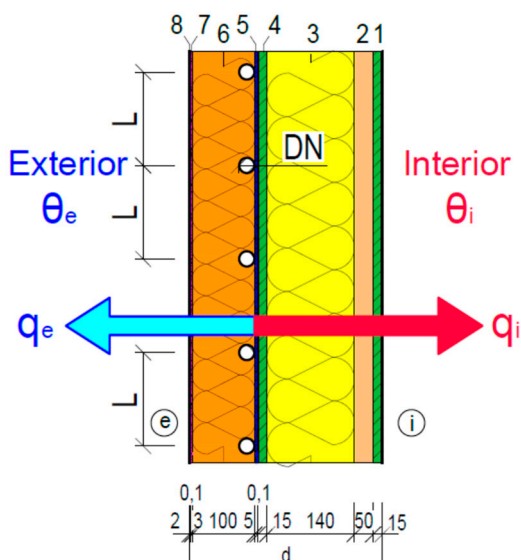

| Mathematical - physical model for TB system | | | | | |
|---|---|---|---|---|---|
| Number | Name of the material | Thickness | Volumetic weight | Thermal conductivity coefficient | Specific heat capacity |
| | Symbol | d | ρ | λ | c |
| | Unit | m | kg/m³ | W/(m·K) | J/(kg·K) |
| 1 | Plasterboard | 0.015 | 840 | 0.142 | 960 |
| 2 | Wood batten | 0.050 | 600 | 0.22 | 2510 |
| 3 | Thermal insulation EPS | 0.140 | 21 | 0.033 | 840 |
| 4 | Plasterboard | 0.015 | 840 | 0.142 | 960 |
| 5 | Adhesive spatula | 0.005 | 1300 | 0.800 | 1020 |
| 6 | Thermal insulation EPS | 0.100 | 100 | 0.035 | 1020 |
| 7 | Reinforced mortar | 0.003 | 1300 | 0.800 | 1020 |
| 8 | Exterior plaster | 0.002 | 1800 | 0.800 | 920 |

**Figure 6.** Mathematical-physical model of the simulation fragment. $q_e$—radiant flux density towards the exterior (W/m²), $q_i$—radiant flux density towards the interior (W/m²), $\theta_e$—outdoor design temperature in winter (°C), $\theta_i$—internal design temperature (°C), d—construction thickness (mm), DN—pipe dimension (mm), e—exterior, and i—interior.

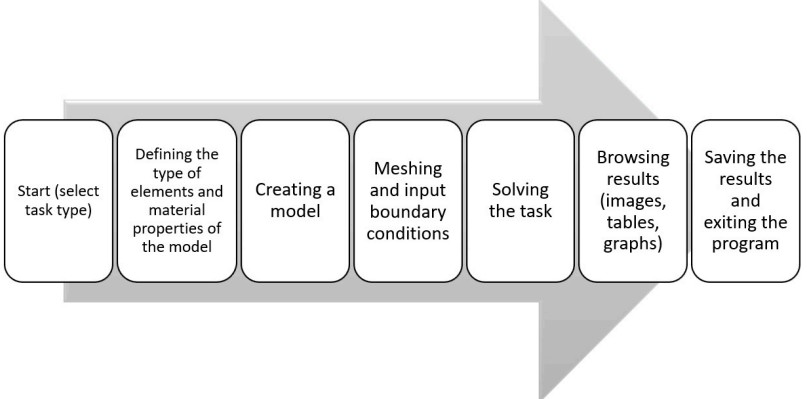

**Figure 7.** The procedure for solving a computer simulation in ANSYS.

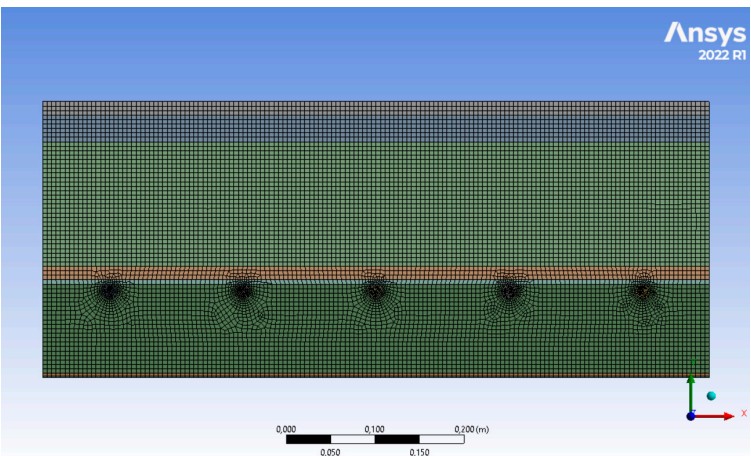

**Figure 8.** Model of a fragment of a structure with a generated mesh.

The Newtonian boundary conditions, i.e., the convection at the boundary with the outside and inside air, are determined by the relations [30]:

$$q_i = h_i \times (\theta(x,y) - \theta_i) = \lambda(x,y) \times \left( \frac{\partial\theta(x,y)}{\partial x} + \frac{\partial\theta(x,y)}{\partial x} \right), \text{ for } x \in \Gamma_I \tag{21}$$

$$q_e = h_e \times (\theta(x,y) - \theta_e) = -\lambda(x,y) \times \left( \frac{\partial\theta(x,y)}{\partial x} + \frac{\partial\theta(x,y)}{\partial x} \right), \text{ for } x \in \Gamma_E \tag{22}$$

where $q_i$ is the radiant flux density towards the interior (W/m$^2$), $q_e$ is the radiant flux density towards the exterior (W/m$^2$), $h_i$ is the heat transfer coefficient from the indoor air to the structure ($h_i = 6$ W/(m$^2$·K)), $h_e$ is the heat transfer coefficient from the structure to the outdoor air ($h_e = 25$ W/(m$^2$·K)), $\theta_i$ is the temperature of the indoor ambient air ($\theta_i = 20$ °C), $\theta_e$ is the temperature of the outdoor ambient air ($\theta_e = -11$ °C), $\theta(x,y)$ is the temperature is a function of two variables, $\lambda(x,y)$ is the coefficient of thermal conductivity is for each of the isotropic materials forming the region (W/(m·K)), and $\Gamma_I, \Gamma_E$ is the boundary of the region on the interior/exterior side.

The task was specified as a thermal task, "Transient Thermal", for the calculation of temperatures in a fragment with a thermal barrier, where a two-dimensional non-stationary heat conduction equation was applied to the given region (specified wall structure). The mathematical relationship applies as follows [30]:

$$c \times \varrho \times \frac{\partial\theta(x,y)}{\partial t} = \left( \frac{\partial}{\partial x}, \frac{\partial}{\partial y} \right) \times \left( \lambda(x,y) \times \left( \frac{\partial\theta(x,y)}{\partial x}, \frac{\partial\theta(x,y)}{\partial y} \right) \right) \tag{23}$$

where c is the specific heat capacity of the material (J/(kg·K), $\varrho$ is the volume mass of the material (kg/m$^3$), $\theta(x,y)$ is the temperature is a function of two variables, t is the time (s), x,y are the variables, and $\lambda(x,y)$ is the coefficient of thermal conductivity is for each of the isotropic materials forming the region (W/(m·K)).

## 4. Results

We summarize the results of our research on the analysis of active thermal protection as a function of the thermal barrier in two points:

- The results of a parametric study of four fragments of building envelopes;
- The results of the computer simulation of the temperature progression in the ATP layer for a fragment of the building envelope of a prefabricated timber building.

### 4.1. Parametric Study of Fragments of Building Envelope Structures

Our research aims to determine the dynamic thermal resistance as a function of static/dynamic thermal insulation thickness and mean temperature of the heat transfer medium in the pipes of the ATP (as a function of TB) heat exchange surface using a parametric study of several material different compositions of fragments of building envelope structures based on the variation of input data of physical variables. At the same time, we determined the potential for energy savings and accumulation and static thermal insulation savings and predicted the application of building structure designs with ATP to the heating/cooling function.

We have developed mathematical-physical models for these types of building envelopes to determine the energy saving and energy storage potential of ATP, as well as to define the dynamic thermal resistance using a parametric study:

- Fragment 1—construction of the ISOMAX system perimeter wall (thermal insulation–reinforced concrete–thermal insulation);
- Fragment 2—reinforced concrete wall with thermal insulation on the exterior side;
- Fragment 3—a wall made of aerated concrete blocks with thermal insulation on the exterior side;
- Fragment 4—prefabricated timber building wall.

These are essentially technical details of the individual fragments with precise information on the building materials (thermal properties and dimensions) of which the envelope is composed. The basis for the determination of the dynamic thermal resistance is the

calculation and progression of the temperatures in the individual layers of the fragments under consideration.

The basic input parameters for the calculation of the internal temperature progression in the individual layers of the building envelope fragments $\theta_m$ (°C) were the exterior temperature $\theta_e = -11$ °C and the interior temperature $\theta_i = +20$ °C.

### 4.1.1. Fragment 1—Construction of the ISOMAX System Perimeter Wall

The first considered Fragment 1 is the construction of the building envelope structure according to the ISOMAX system. In our research, we were inspired by this system. In 2005, we were approached by the ISOMAX system licensee in Slovakia to design, project, and manage the construction of a prototype prefabricated house IDA I (named IDA after the name of the client's wife). This prototype building is built on the premises of a manufacturer of reinforced concrete components for the construction industry in Bratislava-Vrakuňa and is currently used as an office building. We have described the design, project and construction in more detail in the articles [2,31, 32, 33]. Figure 9 is a photograph of the construction of the building. Figure 10 is a photograph of the individual reinforced concrete panels with integrated ATP piping.

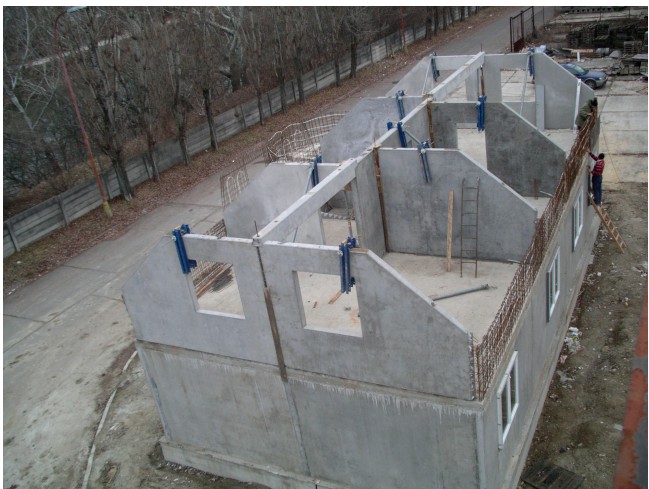

**Figure 9.** View of the assembly of the prototype prefabricated house IDA I. (Photo archive: Kalús).

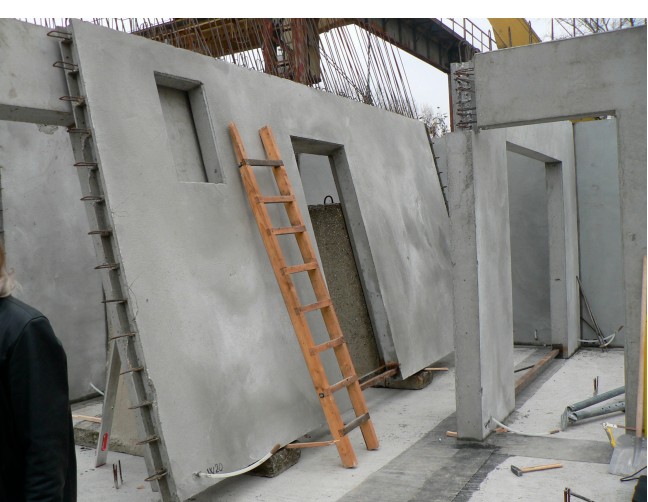

**Figure 10.** View of reinforced concrete panels with integrated ATP piping. (Photo archive: Kalús).

In the ISOMAX system, the load-bearing central structure is made of 150 mm thick reinforced concrete, which is thermally insulated internally and externally with 75 mm thick thermal insulation, Figure 11. Our upgraded design of this envelope retains the 150

mm thickness of the reinforced concrete central load-bearing section but changes the thickness of the interior thermal insulation to 100 mm and the exterior thermal insulation to 200 mm, Figure 12.

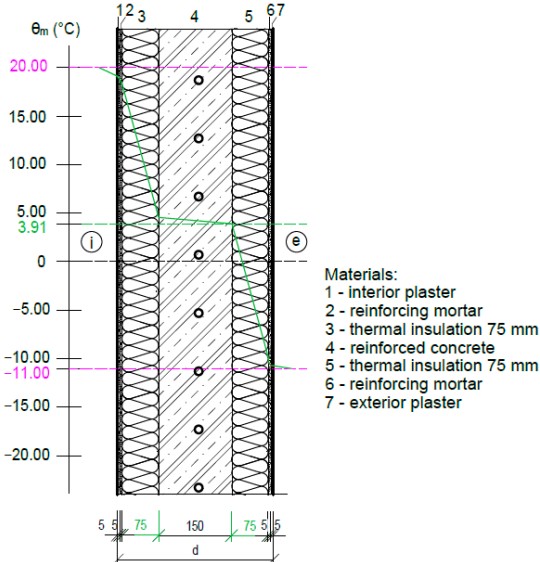

**Figure 11.** Construction of the perimeter wall according to the ISOMAX system [5]. $\theta_m$—the temperature in construction (°C), d—construction thickness (mm), e—exterior, and i—interior.

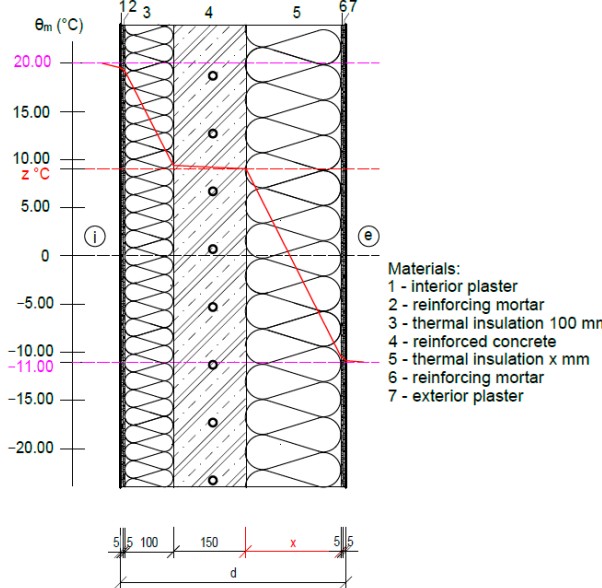

**Figure 12.** Our upgraded ISOMAX perimeter wall construction. $\theta_m$—the temperature in construction (°C), d—construction thickness (mm), x—thickness of thermal insulation, z—the temperature between the load-bearing and thermal insulation layer of the structure (°C), e—exterior, and i—interior.

The calculated internal temperature in the ATP placement layer is $\theta_m$ = 3.91 °C for the ISOMAX building envelope structure design for the static thermal insulation with a thickness of 75 mm, Figure 11, and $\theta_m$ = 9.05 °C for the upgraded design for the static thermal insulation with thickness 200 mm, Figure 12. The dynamic thermal resistance as a function of the thickness of the static/dynamic thermal insulation and the mean temperature of the heat transfer medium in the ATP tubes forming the heat transfer layer in the building structure can be seen from the graph in Figure 13. The figure shows that the standard DTR $R_{DTR}$ = 6.5 ((m²·K)/W) is achieved in this design by the mean temperature of the heat transfer medium in the ATP layer $\theta_m$ = 7.04 °C, and the DTR increases with increasing temperature. A temperature of $\theta_m$ = 16.96 °C represents an $R_{DTR}$ = 29.86 ((m²·K)/W) and an

equivalent static thermal insulation thickness of 1000 mm. Similarly, Figure 14 shows the dependence of the dynamic heat transfer coefficient $U_D$ (W/(m²·K)).

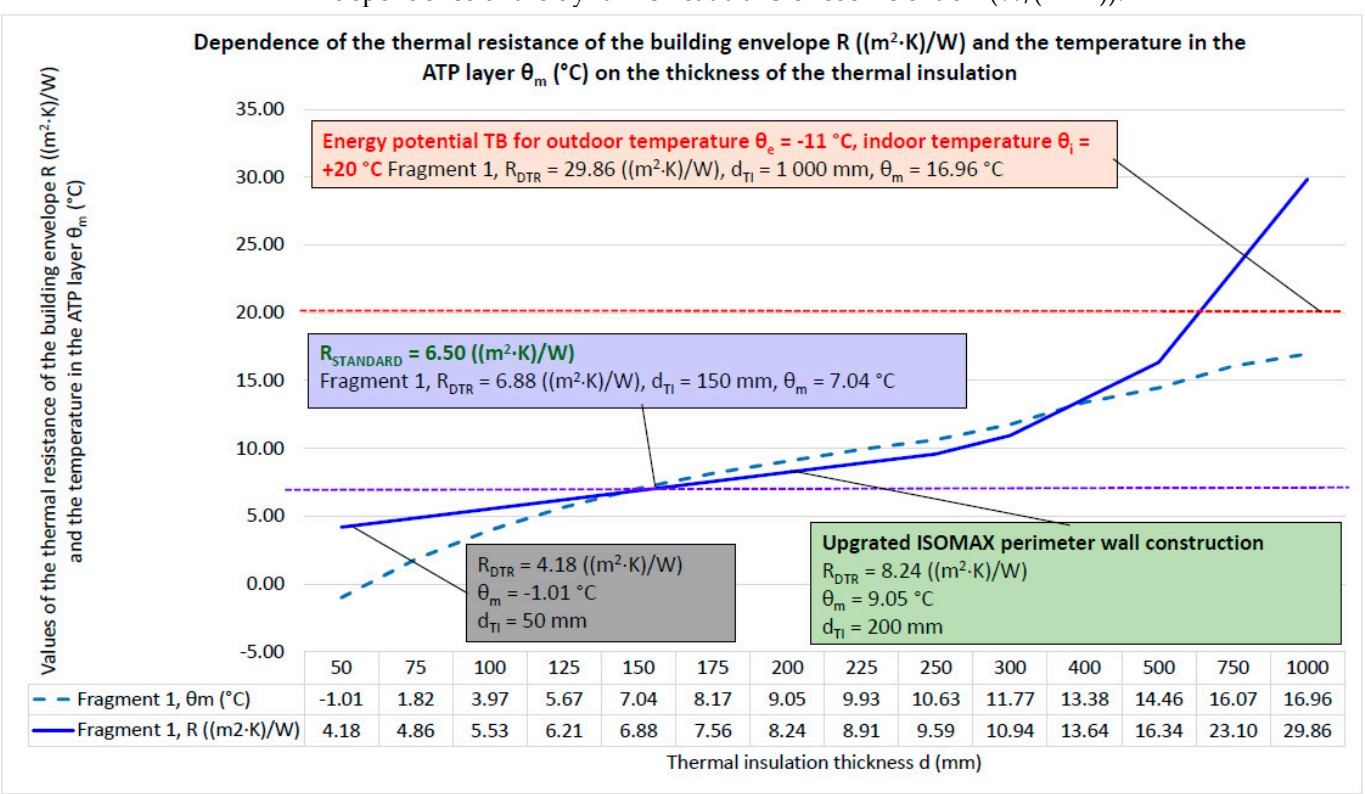

**Figure 13.** Dynamic thermal resistance as a function of static/dynamic thermal insulation thickness and mean temperature of the heat transfer medium in ATP pipes—Fragment 1.

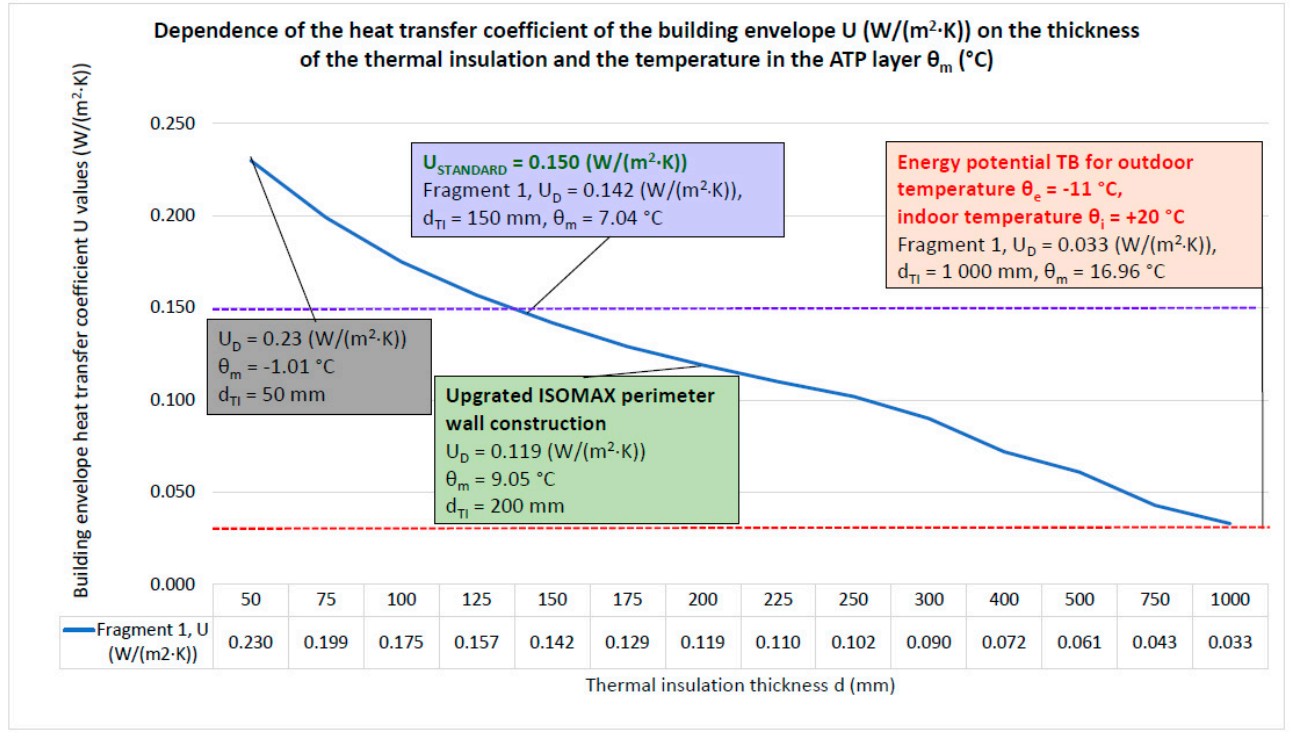

**Figure 14.** Dependence of the dynamic heat coefficient $U_D$ (W/(m²·K)) on the thickness of the thermal insulation and the mean temperature $θ_m$ (°C) in the ATP layer—Fragment 1.

Figure 15 shows in blue the energy saving and energy storage potential of the thermal barrier application for both variants of the envelope design. The energy savings of our upgraded solution relative to the original ISOMAX technical solution are shown in red. Due to the application of thermal insulation also on the interior side, the function of ATP for this building envelope solution is limited to the thermal barrier and heat/cool accumulation functions only.

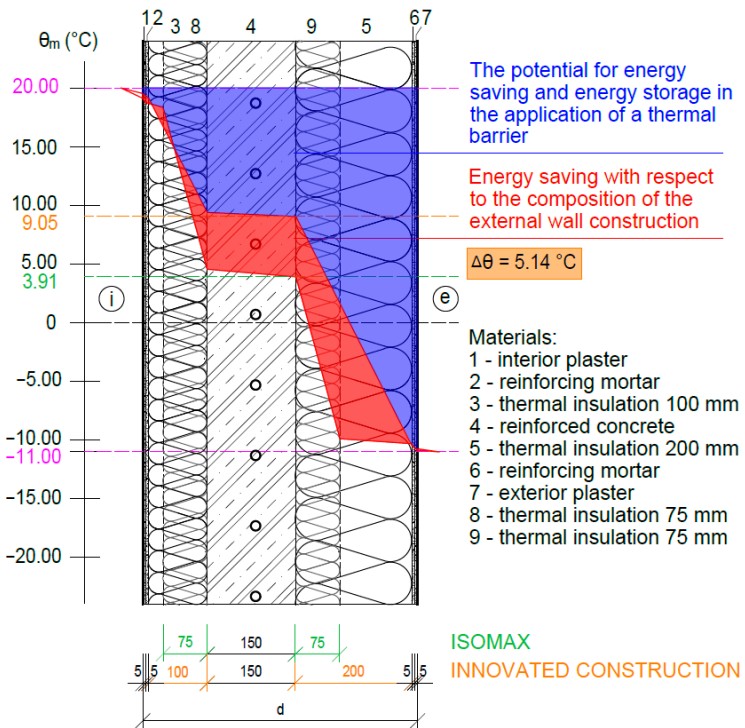

**Figure 15.** Energy saving and energy storage potential of thermal barrier application for both variants of envelope construction. $\theta_m$—the temperature in construction (°C), $\Delta\theta$—temperature difference (°C), d—construction thickness (mm), e—exterior, and i—interior.

Unfortunately, after the construction of the prototype of the prefabricated house IDA I was completed, the owner of the production plant changed, which did not allow us to carry out the planned experimental measurements and verification of the theoretical results obtained based on the parametric study.

### 4.1.2. Fragment 2—Reinforced Concrete Wall with Thermal Insulation on the Exterior Side

The next considered Fragment 2 is a reinforced concrete wall 200 mm thick with thermal insulation on the outside. The parametric study was carried out for a static thermal insulation thickness of 210 mm to achieve the standard thermal resistance and vary the static thermal insulation thickness from 50 mm.

We prepared this fragment for research either in the climate chamber or in the actual construction of the building. In paper [1, 34], we analyzed its energy potential for large-scale radiant heating/cooling with a parametric study. In this section, we analyze ATP in the function of a thermal barrier.

The calculated internal temperature in the ATP placement layer for a thermal insulation thickness of 210 mm is $\theta_m = 18.7$ °C, $R_{DTR} = 6.55$ ((m²·K)/W), Figure 16, and for a thermal insulation thickness of 50 mm is $\theta_m = 15.28$ °C, $R_{DTR} = 1.67$ ((m²·K)/W), Figure 17.

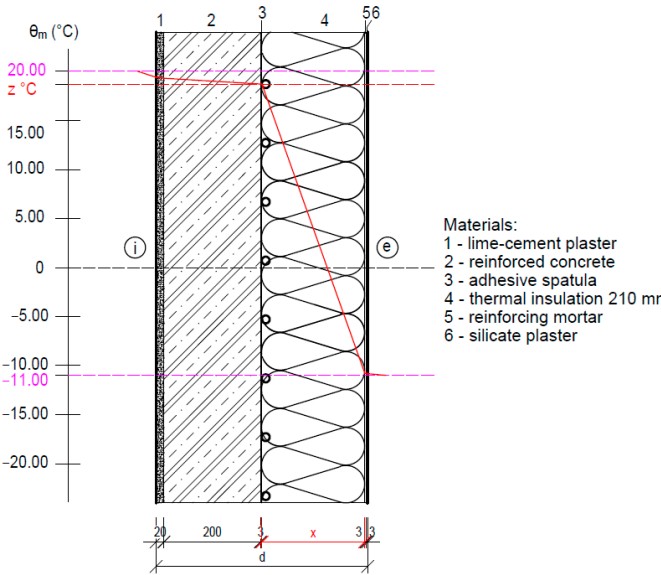

**Figure 16.** Reinforced concrete wall 200 mm thick with thermal insulation 210 mm thick on the outside. $\theta_m$—the temperature in construction (°C), d—construction thickness (mm), x—thickness of thermal insulation, z—the temperature between the load-bearing and thermal insulation layer of the structure (°C), e—exterior, and i—interior.

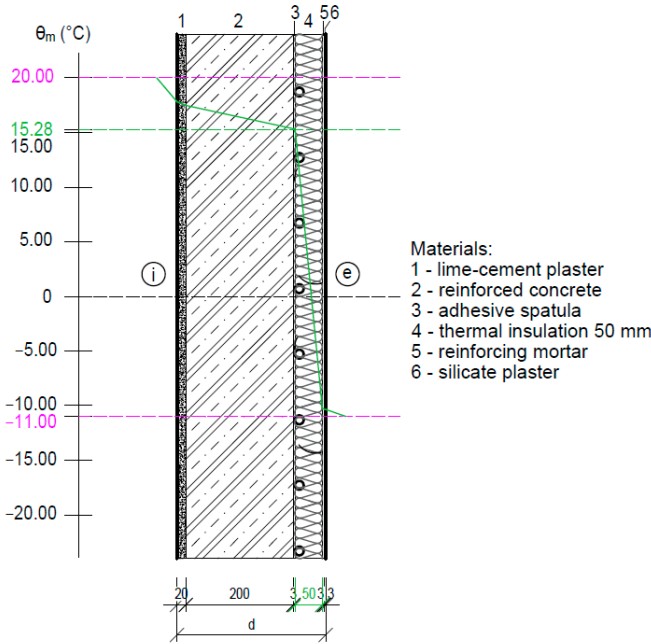

**Figure 17.** Reinforced concrete wall 200 mm thick with thermal insulation 50 mm thick on the outside. $\theta_m$—the temperature in construction (°C), d—construction thickness (mm), e—exterior, and i—interior.

The dynamic thermal resistance as a function of the thickness of the static/dynamic thermal insulation and the mean temperature of the heat transfer medium in the ATP pipes that form the heat exchange layer in the building structure can be seen from the graph in Figure 18. For a mean temperature of the heat transfer medium in the ATP layer of $\theta_m = 19.72$ °C, $R_{DTR} = 30.46$ ((m²·K)/W), an equivalent static thermal insulation thickness of 1000 mm would be required.

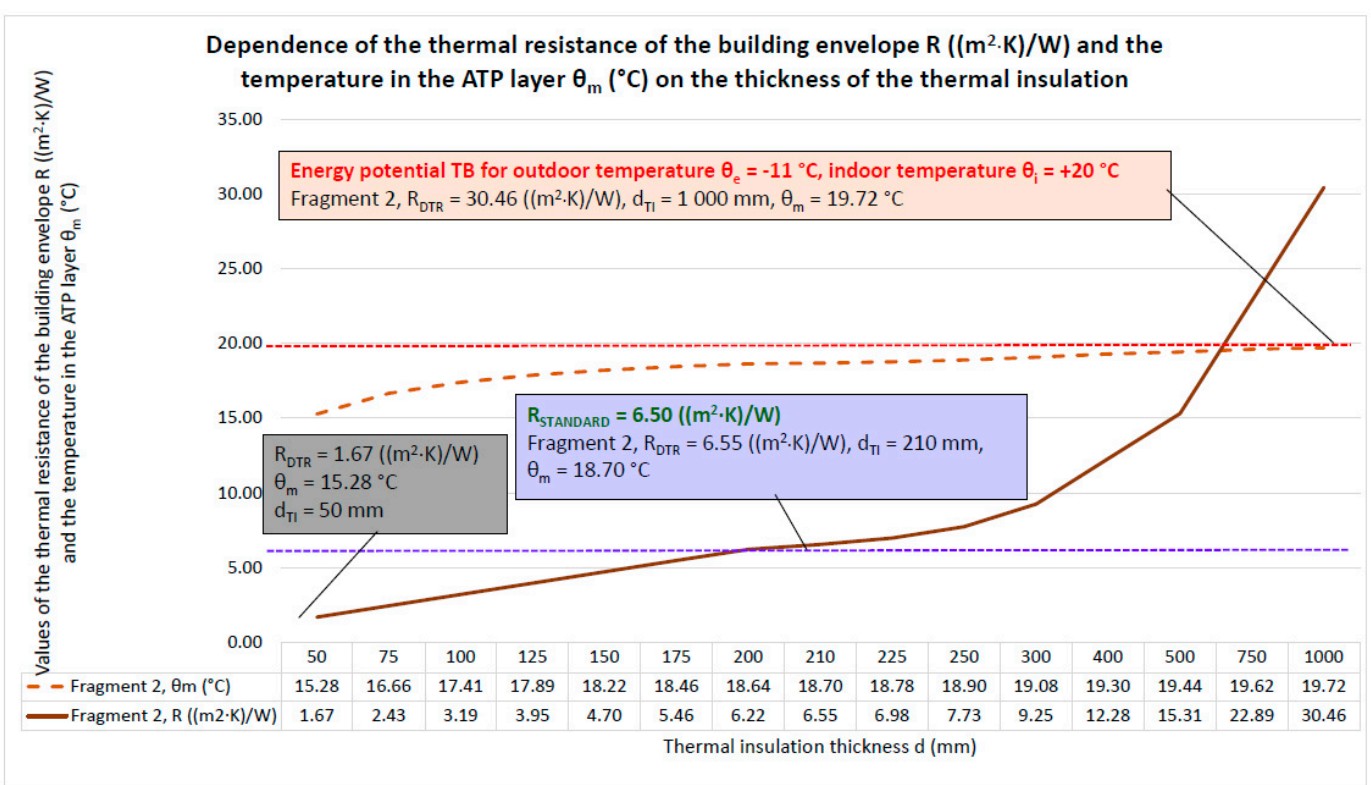

**Figure 18.** Dynamic thermal resistance as a function of static/dynamic thermal insulation thickness and mean temperature of the heat transfer medium in ATP pipes—fragment 2.

Similarly, Figure 19 shows the dependence of the dynamic heat transfer coefficient $U_D$ (W/(m²·K)). Figure 20 shows in red the required heat in kWh delivered by a heat transfer medium with a mean temperature of 18.7 °C to the ATP (function TB) when using a 50 mm thick static thermal insulation to bring the dynamic thermal resistance of the building envelope up to the value of the standard thermal resistance when using a 210 mm thick static thermal insulation. Based on the results of the study [34], we can conclude that with this technical solution of the building envelope, it is possible to obtain savings on the thickness of the static thermal insulation up to 160 mm. According to the results of the study [1], the ATP function for this building envelope solution has a high potential for application as wall heating/cooling in addition to the thermal barrier and heat/cooling storage functions.

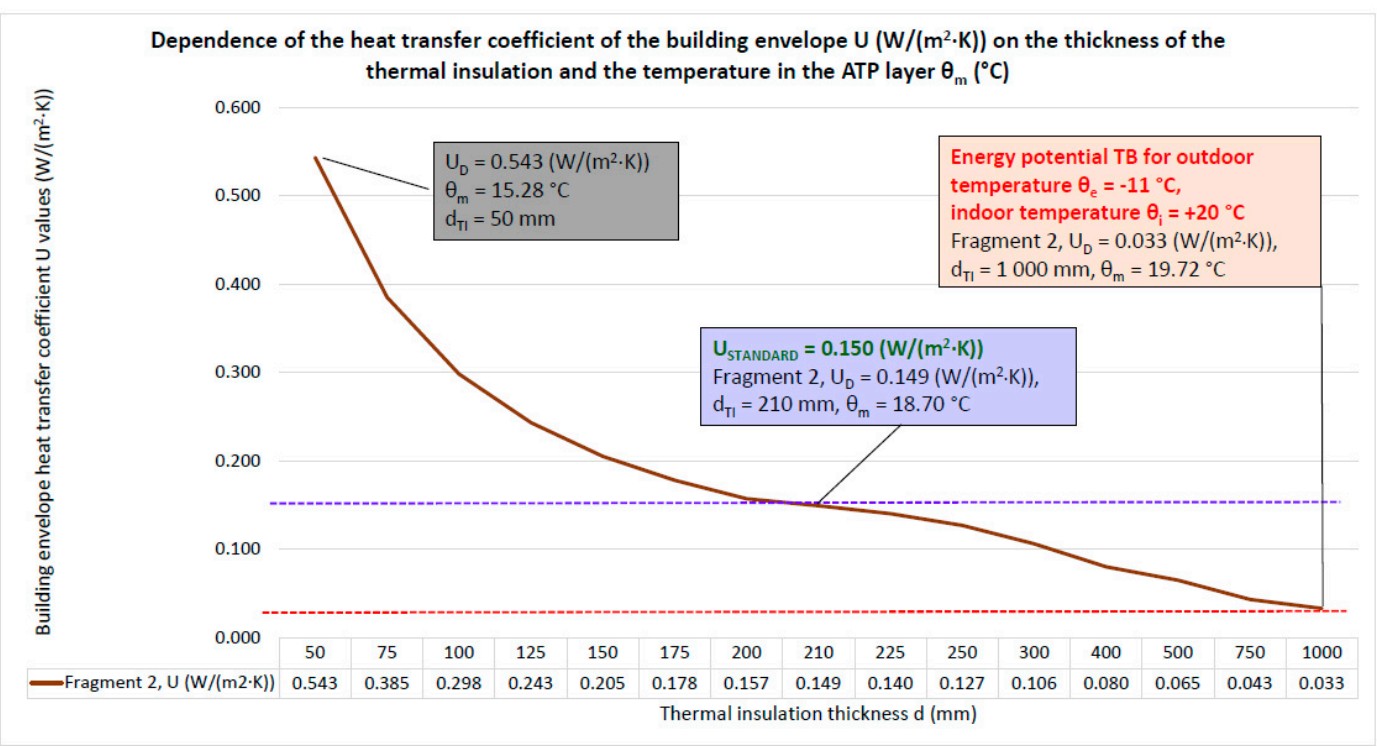

**Figure 19.** Dependence of the dynamic heat coefficient UD (W/(m2·K)) on the thickness of the thermal insulation and the mean temperature θm (°C) in the ATP layer—Fragment 2.

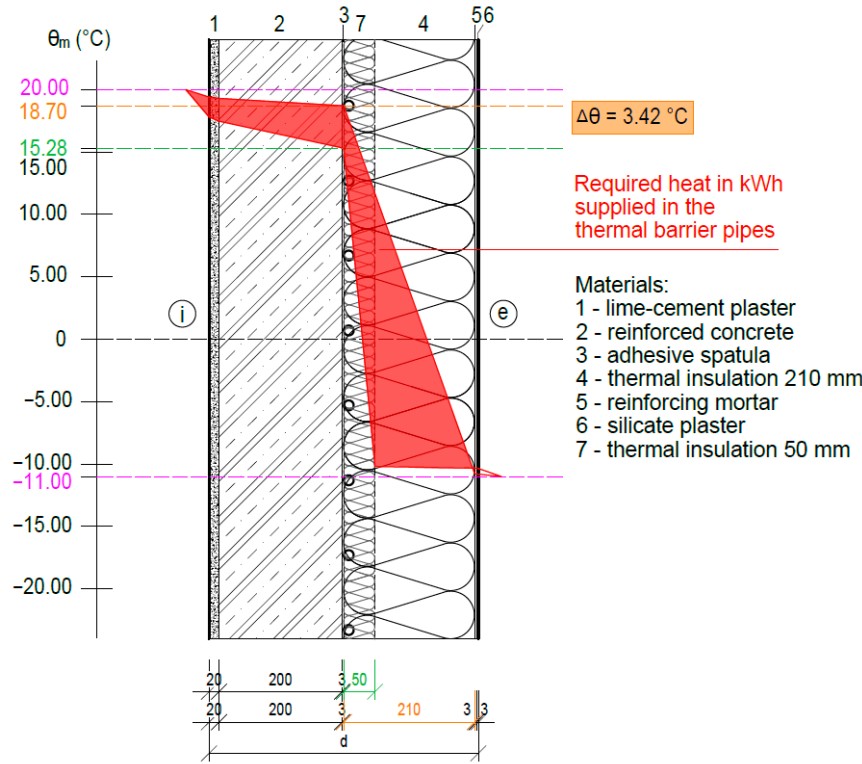

**Figure 20.** Required heat in kWh supplied by a heat transfer medium with a mean temperature of 18.7 °C to the ATP (function TB) using 50 mm thick static thermal insulation to achieve the standard thermal resistance. θm—the temperature in construction (°C), Δθ—temperature difference (°C), d—construction thickness (mm), e—exterior, and i—interior.

### 4.1.3. Fragment 3—A Wall Made of Aerated Concrete Blocks with Thermal Insulation on the Exterior Side

Then, we analyze Fragment 3—a wall with a load-bearing part made of aerated concrete blocks with a thickness of 375 mm and thermal insulation from the outside with a thickness of 100 mm. We applied this type of fragment to an experimental family house EB2020 in Tomášov near Bratislava, Slovakia, which we designed, projected, managed its construction, and conducted experimental measurements on it. We carried out these activities between 2010 and 2013.

Between the facade's polystyrene (100 mm) and aerated concrete masonry (375 mm) and in the roof structure's succeeding circuits, the building's active thermal protection is provided by plastic piping: 20 m × 100 m. Using the ATP, a building can be heated or cooled during the summer by reducing heat loss via opaque materials. The source of cooling comes from the cooling circuits, which are buried in the ground near the foundation strips of the building at a non-freezing depth. They consist of plastic pipe circuits: 20 m × 100 m. The ATP is shown in Figure 21.

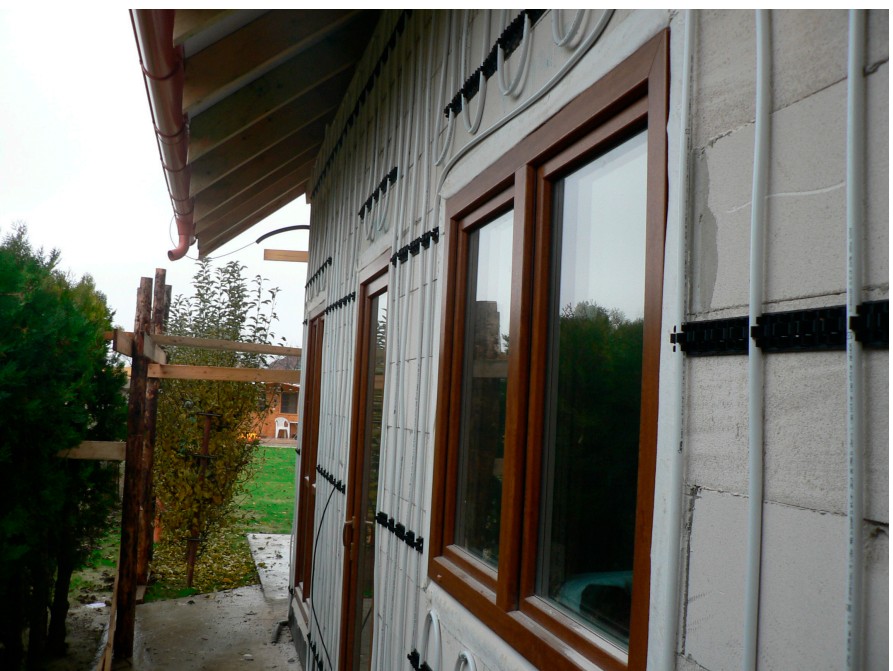

**Figure 21.** View of the ATP formed by a plastic pipe between aerated concrete masonry and polystyrene. (Photo archive: Kalús).

We have described the experimental measurements and evaluation of the energy roof in [35] and of the ground-source heat storage in [33]. In this paper, we describe the analysis of a parametric study of the application of ATP as a thermal barrier in this type of building envelope.

The calculated internal temperature in the ATP placement layer is $\theta_m$ = 1.89 °C, $R_{DTR}$ = 6.48 ((m²·K)/W), Figure 22. Figure 23 shows the mathematical-physical model for calculating the dynamic thermal resistance for varying thicknesses of static thermal insulation. Figure 24 shows the graphical dependence of the dynamic thermal resistance on the thickness of the static/dynamic thermal insulation and the mean temperature of the heat transfer medium in the ATP tubes that form the heat transfer layer in the building structure. Similarly, Figure 25 shows the dependence of the dynamic heat transfer coefficient $U_D$ (W/(m²·K)). The dynamic thermal resistance of this building envelope for the illustrated isotherm with an internal temperature in the ATP location layer of $\theta_m$ = 10 °C is $R_{DTR}$ = 11.8 ((m²·K)/W), the thickness of 300 mm thermal insulation, respectively, for $\theta_m$ = 16.11 °C is $R_{DTR}$ = 30.80 ((m²·K)/W), the thickness of 1000 mm thermal insulation, Figure 26. Because

the load-bearing wall made of porous concrete blocks has a high thermal resistance, the function of the ATP for this building envelope solution is limited only to the functions of thermal barrier and partial heat/cold accumulation.

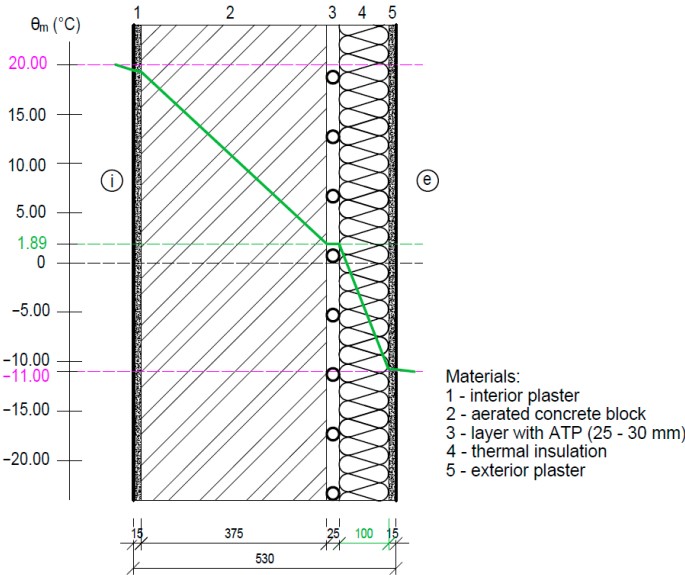

**Figure 22.** Fragment 3—wall with a load-bearing part made of aerated concrete blocks with a thickness of 375 mm and thermal insulation on the outside with a thickness of 100 mm. $\theta_m$—temperature in construction (°Ce—exterior, and i—interior.

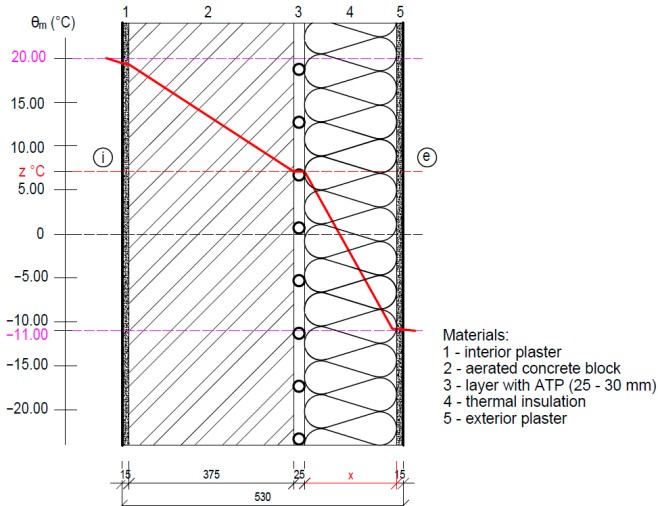

**Figure 23.** The mathematical-physical model for calculating the dynamic thermal resistance for varying thicknesses of static thermal insulation. $\theta_m$—the temperature in construction (°C), x—thickness of thermal insulation (mm), z—the temperature between the load-bearing and thermal insulation layer of the structure (°C), e—exterior, and i—interior.

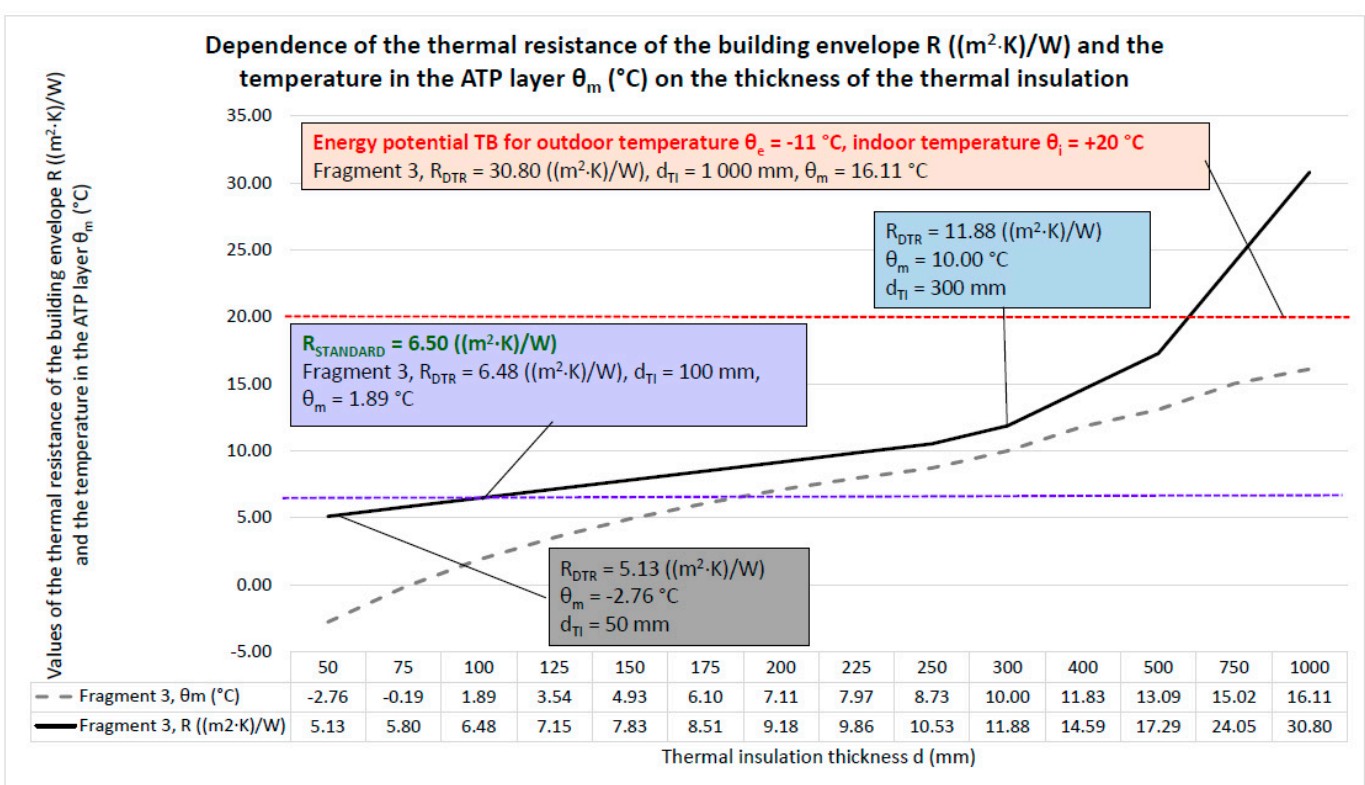

**Figure 24.** Graphical dependence of the dynamic thermal resistance on the thickness of the static/dynamic thermal insulation and the mean temperature of the heat transfer medium in the ATP pipes—Fragment 3.

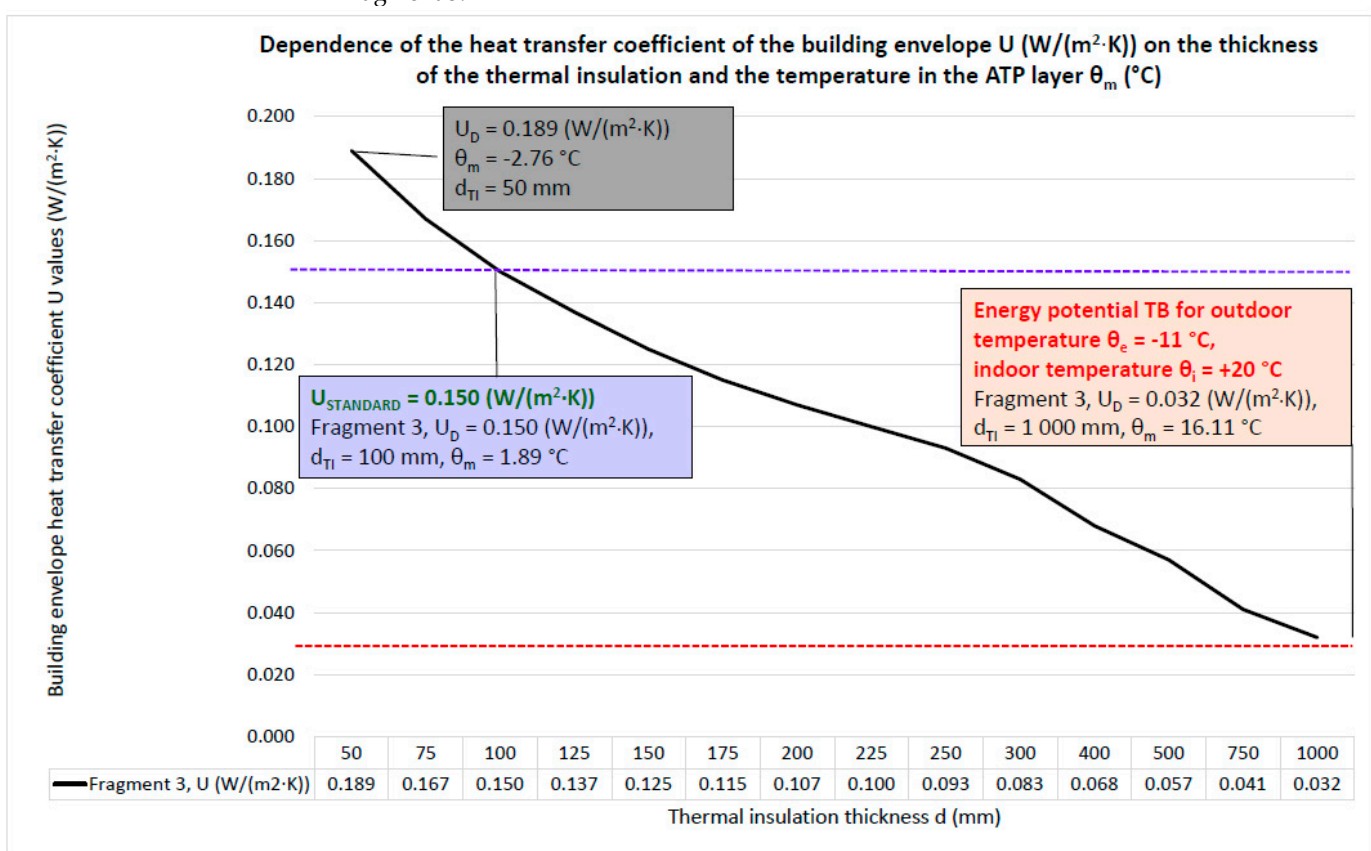

**Figure 25.** Dependence of the dynamic heat coefficient $U_D$ (W/(m²·K)) on the thickness of the thermal insulation and the mean temperature $\theta_m$ (°C) in the ATP layer—Fragment 3.

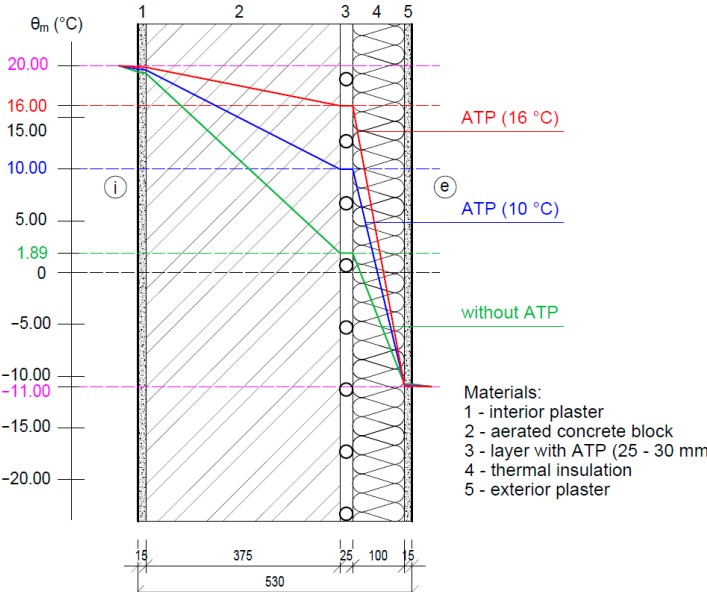

**Figure 26.** View of the ATP formed by a plastic pipe between aerated concrete masonry and polystyrene. $\theta_m$—the temperature in construction (°C).

### 4.1.4. Fragment 4—Prefabricated Timber Building Wall

The last analyzed building envelope is Fragment 4—the wall of the prefabricated timber building. The variant construction of the perimeter wall consists of plasterboard, thermal insulation with a thickness of 50 mm, OSB board, vapor barrier, thermal insulation of 140 mm, OSB board, thermal insulation of 160 mm, reinforced mortar, and exterior plaster. Figure 27 shows two options for the location of active thermal protection for the composition of these structures and the temperature progression in this structure without the application of ATP.

Considering the requirements of practice and the demand for the construction of prefabricated wooden buildings, we are preparing research on the application of ATP in the function of a thermal barrier for such buildings. To verify the dynamic thermal resistance for a fragment of the perimeter wall of a timber building, we designed a test cell, which is under construction at the time, Figure 28. The load-bearing part of the structure is formed by the load-bearing columns of the wooden perimeter wall of the test cell with dimensions 60 × 140 mm filled with 140 mm thick thermal insulation. On the exterior side, the test cell will be insulated with self-made prototypes of thermal insulation panels with integrated energy-active elements—active thermal protection (ATP) with a thickness of 100 mm, Figure 29.

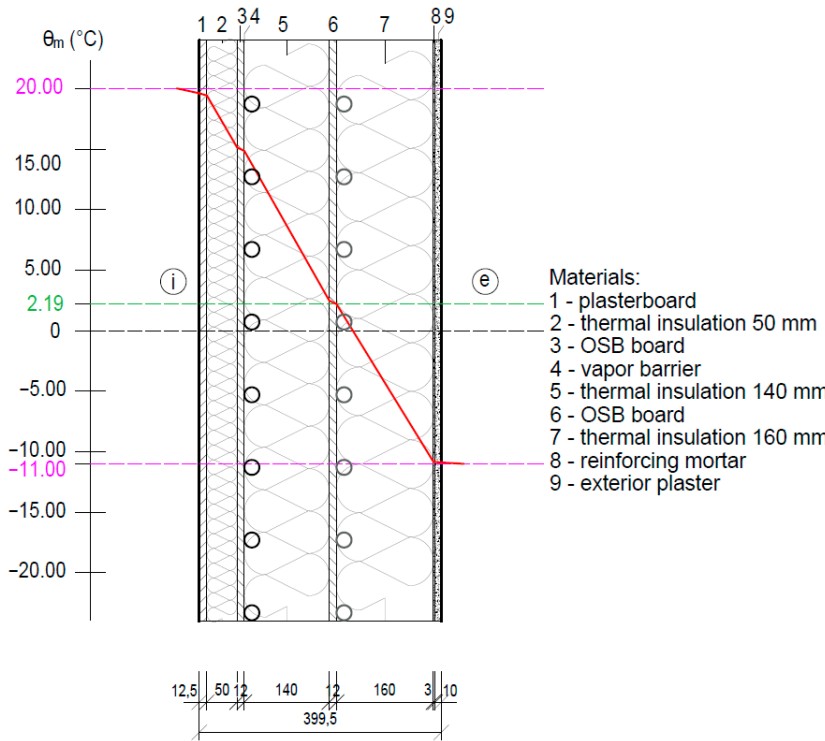

**Figure 27.** Temperature behavior in the construction of a prefabricated timber house. $\theta_m$—temperature in construction (°C), e—exterior, and i—interior.

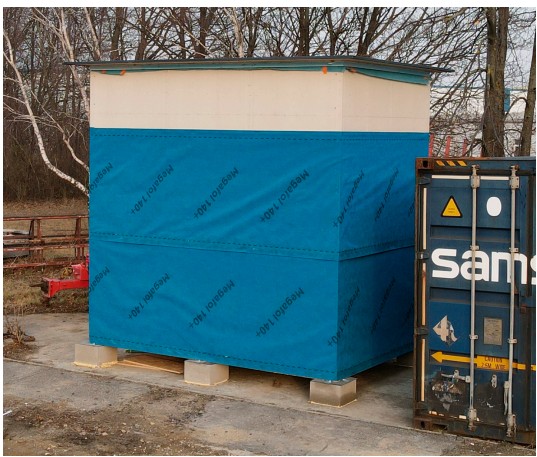

**Figure 28.** View of a test cell under construction. (Photo archive: Ingeli).

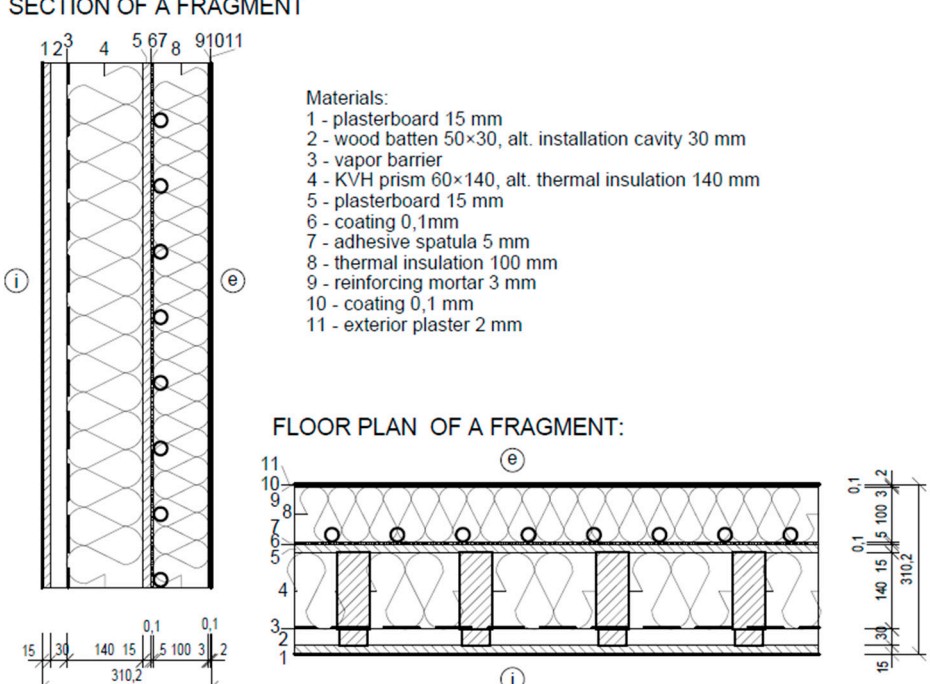

**Figure 29.** Details of test cell wall fragments. e—exterior, and i—interior.

To analyze the dynamic thermal resistance of this wall structure, we developed a mathematical-physical model, Figure 30. The calculated internal temperature in the ATP placement layer for a thermal insulation thickness of 200 mm is $\theta_m$ = 6.03 °C, Figure 31, and for a thermal insulation thickness of 100 mm is $\theta_m$ = 0.80 °C. If we increase the mean temperature in the ATP heat transfer layer by $\Delta\theta$ = 5.23 °C using a heat transfer agent, the dynamic thermal resistance is $R_{DTR}$ = 10.487 ((m²·K)/W). At the same time, we eliminate the thickness of the static thermal insulation by 100 mm.

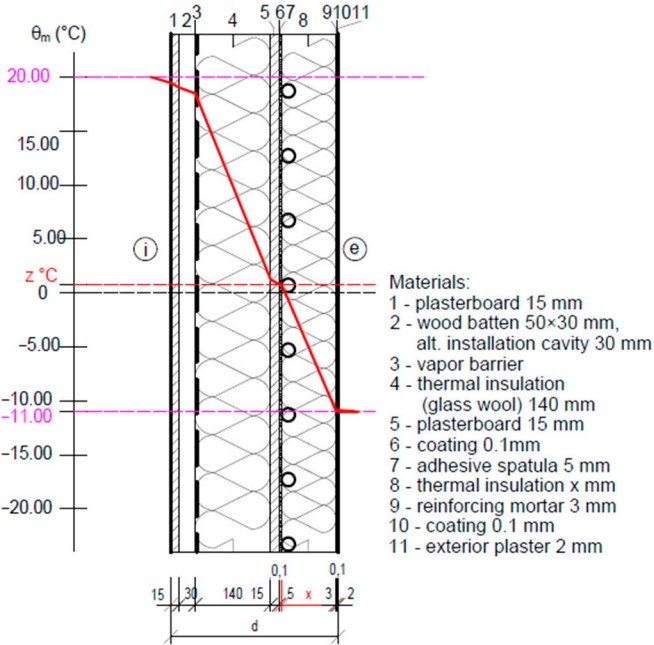

**Figure 30.** Mathematical-physical model of the test cell wall. $\theta_m$—the temperature in construction (°C), d—construction thickness (mm), x—thickness of thermal insulation (mm), z—the temperature between the load-bearing and thermal insulation layer of the structure (°C), e—exterior, and i—interior.

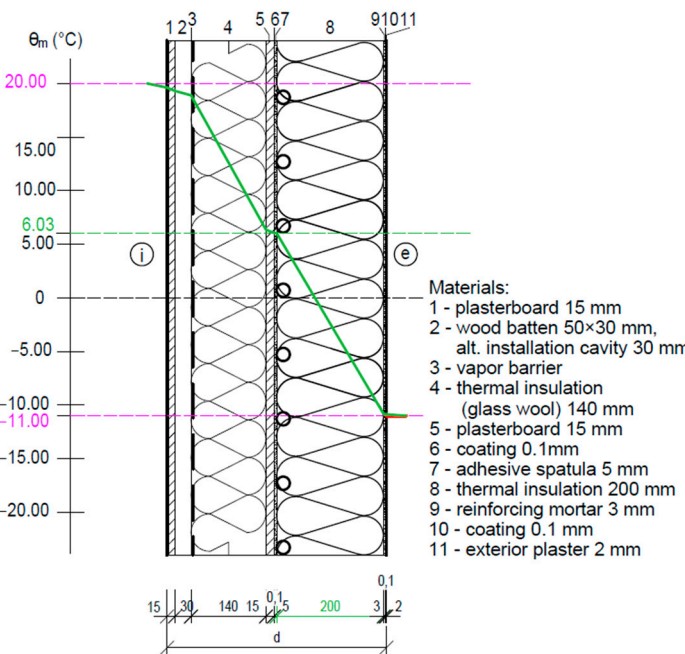

**Figure 31.** Mathematical-physical model of a test cell wall with a static insulation thickness of 200 mm. $\theta_m$—the temperature in construction (°C), d—construction thickness (mm), e—exterior, and i—interior.

The dynamic thermal resistance as a function of the thickness of the static/dynamic thermal insulation and the mean temperature of the heat transfer medium in the ATP tubes, which form the heat transfer layer in the building structure, can be seen from the graph in Figure 32. Similarly, Figure 33 shows the dependence of the dynamic heat transfer coefficient $U_D$ (W/(m²·K)).

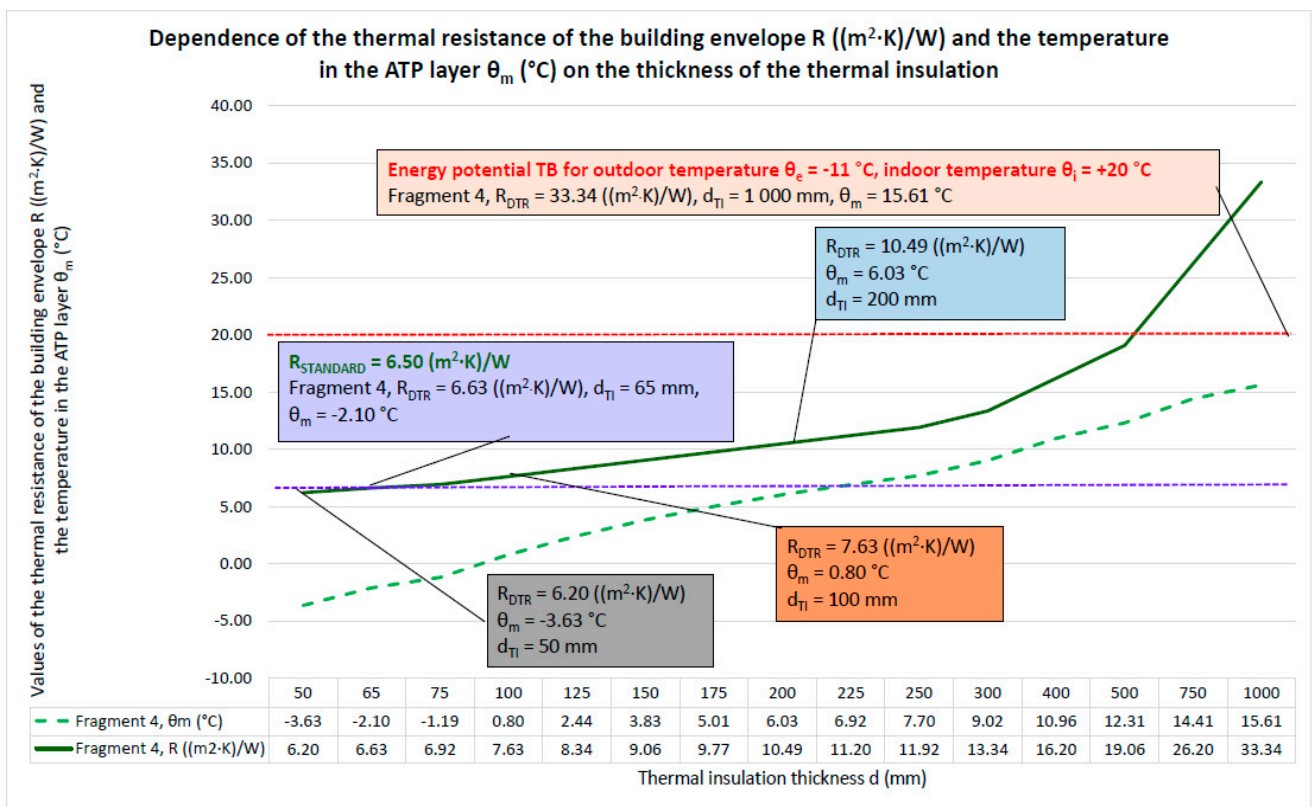

**Figure 32.** Graphical dependence of the dynamic thermal resistance on the thickness of the static/dynamic thermal insulation and the mean temperature of the heat transfer medium in the ATP pipes—Fragment 4.

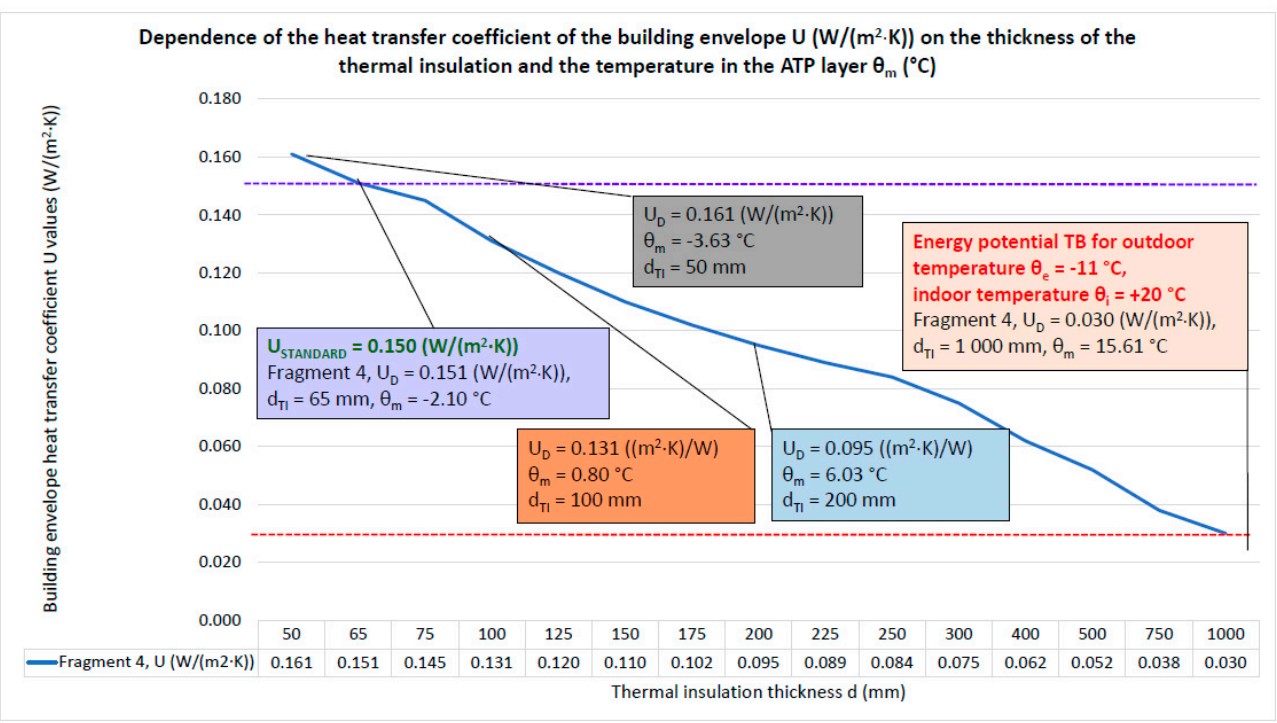

**Figure 33.** Dependence of the dynamic heat coefficient $U_D$ (W/(m²·K)) on the thickness of the thermal insulation and the mean temperature $\theta_m$ (°C) in the ATP layer—Fragment 4.

Figure 34 shows in green the required heat in kWh delivered by a heat transfer medium with mean temperature $\theta_m$ = 6.03 °C to the ATP (function TB) using 100 mm thick static thermal insulation, $\theta_m$ = 0.8 °C, so that the dynamic thermal resistance of the building envelope reaches the value of the thermal resistance corresponding to the use of 200 mm thick static thermal insulation.

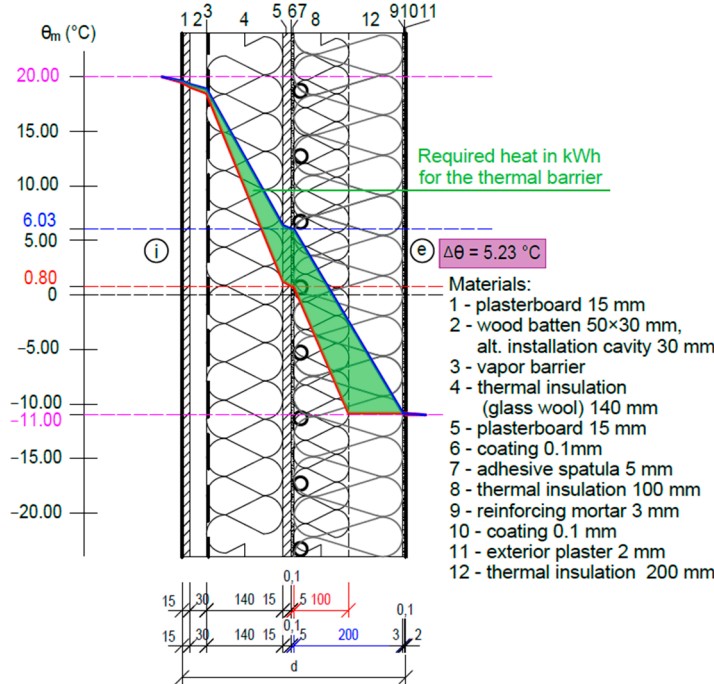

**Figure 34.** Required heat in kWh delivered by the heat transfer medium to the ATP (TB function) to increase the DTR corresponding to a thermal insulation thickness of 200 mm for a static thermal insulation thickness of 100 mm. $\theta_m$—the temperature in construction (°C), $\Delta\theta$—temperature difference (°C), d—construction thickness (mm), e—exterior, and i—interior.

Based on the analysis of the dynamic thermal resistance, we can conclude that in this building envelope design, ATP is only significant as a function of TB. However, at relatively low mean temperatures of the heat transfer medium $\theta_m$ = 15.61 °C, the dynamic thermal resistance has a high value of $R_{DTR}$ = 30.34 ((m²·K)/W), which corresponds to a static thermal insulation thickness of 1000 mm.

*4.2. Computer Simulation of the Progression of the Temperature on a 2D Model of a Fragment of the Perimeter Wall*

Active thermal protection in all its functions and the dynamic thermal resistance of building structures with integrated energy-active elements depend on the uniform and continuous maintenance of the temperature in the heat exchange layer of the building structure in which it is located. For these reasons, the subject of our research is the optimal way to distribute heat in this layer to achieve the desired temperature. For this purpose, we used computer simulation in ANSYS.

The heat exchange surface of the ATP is formed by tubes through which the heat transfer fluid flows. Several technical parameters enter the calculations, such as the dimension of the tubes, the axial distance of the tubes, the thermal-technical properties of the tube material and the layer in which they are placed, the mean temperature of the heat transfer medium, the thermal-technical properties and thicknesses of the building materials forming the building envelope, and the interior and exterior temperature.

The computer simulation of the temperature history on the 2D model was carried out for Fragment 4—the wall of the prefabricated timber building, Figure 30, namely the composition of the building envelope of the test cell. In determining the boundary conditions for the determination of the dynamic thermal resistance, we relied on the data from the parametric study presented in the previous section. A thermal barrier time regime of 27 h was considered in the simulations.

The created 2D model consists of 5 heating pipes, where the heat carrier is water, which has a mean temperature of 6 °C and represents the dynamic thermal resistance corresponding to the standard (static) thermal resistance of the building envelope under consideration. ATP, in this case, has the function of TB. The temperature in the ATP layer at a static thermal insulation thickness of 100 mm is determined by calculation in the parametric study to be 0.8 °C.

Figure 6 in Section 3.3 shows the mathematical-physical model of the fragment that was used for the computer simulation. For the computer simulation, we considered plastic–aluminum pipes. The dimensions of the tubes are 16 × 1.5 mm, the axial distance of the tubes is 150 mm, the thermal conductivity of the material of the pipes $\lambda$ = 0.35 W/(m·K), the thermal conductivity of the layer in which they are placed is $\lambda$ = 0.035 W/(m·K), and the mean temperature of the heat transfer medium is 6 °C. The thermal-technical properties and thicknesses of the building envelope materials are shown directly in Figure 2, and we have considered an internal design temperature of +20 °C and an external design temperature of −11 °C.

After the calculations were performed, the simulation results were generated, namely the temperature waveforms in the structure in the form of a model of the structure, tables, and graphs. The temperature history of the structure can also be displayed in the form of isotherms, Figure 35, and a preferred view can be selected, Figure 36. The figures show a uniform and continuous distribution of temperatures in the different layers of the considered fragment of the building structure. The color range is from a maximum temperature of 20 °C, shown in red, to a minimum temperature of −11 °C, shown in dark blue.

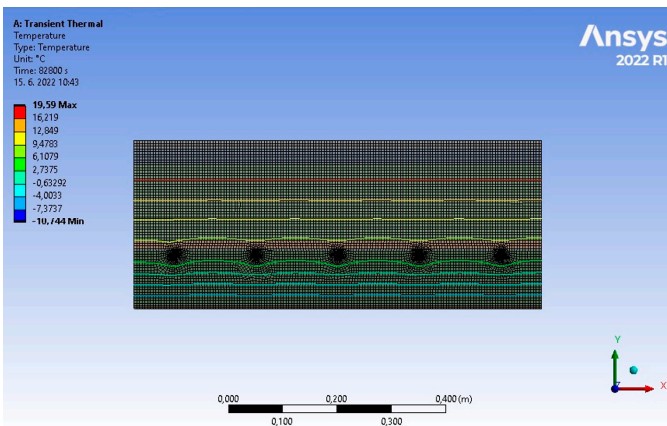

**Figure 35.** Display of results in the form of isotherms.

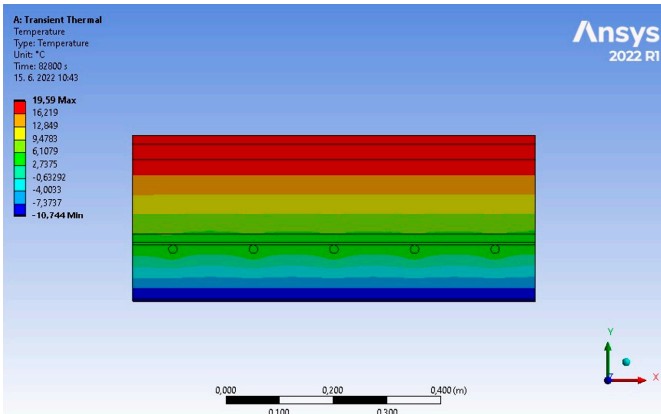

**Figure 36.** Display of the results with the building envelope structure visible.

An important result of the computer simulation is the uniform and continuous temperature distribution in the ATP heat transfer layer with a temperature of about 6 °C, which confirms the functionality of the TB with the achievement of a dynamic thermal resistance $R_{DTR}$ = 10.487 ((m²·K)/W) at a dynamic thermal insulation thickness of 100 mm, which is equal to the thermal resistance at a static thermal insulation thickness of 200 mm.

This computer simulation was intended only for a basic analysis of the uniform and continuous temperature distribution in the ATP layer. We will continue the simulations to analyze for different changes in input parameters, the heat fluxes to the interior and exterior, the amount of heat delivered, the effect of operating time, and other physical variables affecting the dynamic thermal resistance of the individual building envelope structures.

## 5. Discussion

Following the research work reported in Chapter 2 and a selection of the significant outputs of these researchers reported in this chapter, the aim of the research described in this paper was a parametric study of the dynamic thermal resistance of four building envelope fragments, Chapter 4.1, the simulation of the temperature evolution in the individual layers of the fragment—prefabricated timber building wall structure, Chapter 4.2, and the determination of the energy potential of these materially different fragments of building envelope wall structures with integrated energy active elements at different input data of physical quantities. In this section, we present important research outputs from the peer-reviewed articles mentioned in Chapter 2 and describe the published results of our previous research in this area, which form the basis for the analysis of the dynamic

thermal barrier of building envelopes and the research carried out using the parametric study and computer simulation presented in this article.

Koenders, S. J. M., Loonen, R. C. G. M., and Hensen, J. L. M., 2018 [8], based on a simulation model of a new type of closed-loop dynamic insulation system with forced convection, realized that up to nine times lower heat transfer coefficient *U* can be achieved compared to a conventional static insulation system. A schematic diagram of the thermal resistance of a conventional wall and a wall equipped with a dynamic insulation system is shown in Figure 37.

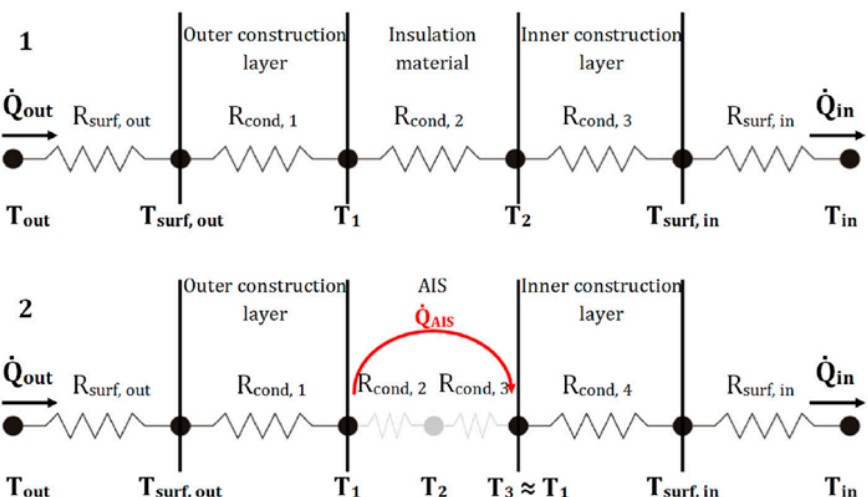

**Figure 37.** Thermal resistance diagram of (**1**) a typical wall and (**2**) a wall equipped with a dynamic insulation system [8]. $R_{cond,x}$—thermal resistance of AIS ((m²·K)/W), $R_{surf}$—thermal resistance of surface ((m²·K)/W), $T_x$—temperature at point x (°C), $Q_x$—radiant flux density (W/m²), and AIS—active insulation system.

Kisilewicz, T., Fedorczak-Cisak, M., and Barkanyi, T. [10], 2019, conducted research in an experimental residential building with an innovative direct-interface ground-source heat exchanger system with dynamic thermal insulation. The first results from measurements and simulations showed a decrease in the thermal losses of the external walls by an average of 63% (53% in November and 81% in March), while the equivalent heat transfer coefficient of the analyzed structure depended on the local climatic conditions. Its value was 0.047 W/(m²·K) in November and 0.11 W/(m²·K) in March for the analyzed wall, while the standard value was 0.282 W/(m²·K).

Fawaier, M. and Bokor, B. [11], 2022, reported that a low to zero heat transfer coefficient (*U* < 0.1) could be achieved by the application of dynamic airflow-based insulation. This could result in energy savings of more than 40% compared to a conventional envelope meeting the prescriptive criteria. They provided a graphical SWOT analysis for the application of dynamic isolation, Figure 38.

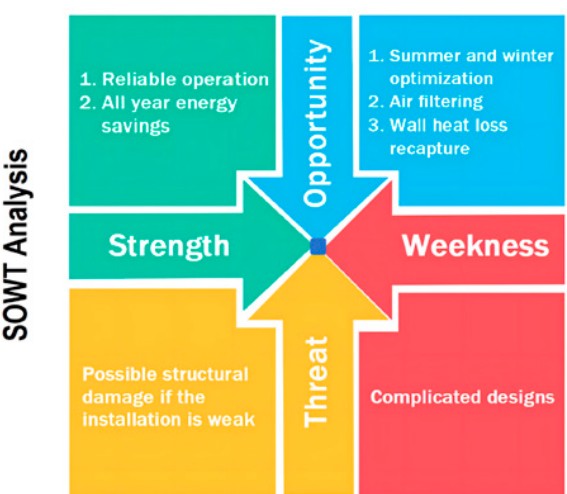

**Figure 38.** SWOT analysis for the dynamic insulation approach [11]. SWOT strengths, weaknesses, opportunities, and threats analysis.

Shen, J., Wang, Z., Luo, Y., Jiang, X., Zhao, H., and Tian, Z. [12], 2022, developed a mathematical model of an active envelope system with redistribution of absorbed solar heat gain between the south and north facades. The thermal performance of the system was simulated under typical weather conditions in five building climate regions of China and then compared with a conventional wall system. The results from the measurements showed that the system with dynamic insulation in the hot summer and cold winter zones reduces heat loss during the heating season by approximately 12.8%. The savings in the very cold climate and moderately cold climate are 4.6% and 8.7%, respectively. Savings are the lowest in the hot summer and warm winter zones.

Chen S., Yang Y., and Chang T. [16], 2023, analyzed the hydraulic thermal barrier (HTB), which enables the building envelope to be gradually viewed as a multifunctional element and provides a chance to change the characteristics of thermal insulation solutions from high-carbon to zero-carbon. To assist in creating a continuous thermal buffer zone within the building envelope, pipe spacing should ideally be between 100 and 250 mm.

The analysis of the dynamic thermal barrier using a parametric study of four materially different constructions fragments of the building envelope of the most widely used in practice confirmed the results presented in articles by the cited researchers. The results of the dependencies of the dynamic thermal resistance and the dynamic heat transfer coefficient for the fragments studied are presented in the graphs in Figures 39 and 40. Figure 39 shows the graphical dependencies of the dynamic thermal resistance $R_{DTR}$ ((m²·K)/W) on the thickness of the static/dynamic thermal insulation and the mean temperature of the heat transfer medium in the ATP tubes for all fragments investigated. The relatively low mean temperature of the heat transfer medium $\theta_m$ (°C) = 15.61 to 19.72 °C delivered to the tubes of the ATP heat transfer layer gives a dynamic thermal resistance of $R_{DTR}$ = 29.86 to 33.34 ((m²·K)/W) with an equivalent dynamic thermal insulation thickness of 1,000 mm for the required standard resistances $R_{STANDARD}$ = 6.50 ((m²·K)/W) of the individual fragments of the building envelope with static thermal insulation of 65 to 210 mm. Then, the energy potential of using TB is 455 to 513% in thermal resistance and 476 to 1.538% in thickness dynamic thermal insulation. Figure 40 shows the graphical dependencies of the dynamic heat transfer coefficient $U$ (W/(m²·K)) on the thickness of the static/dynamic thermal insulation and the mean temperature of the heat transfer medium in the ATP tubes for all fragments investigated.

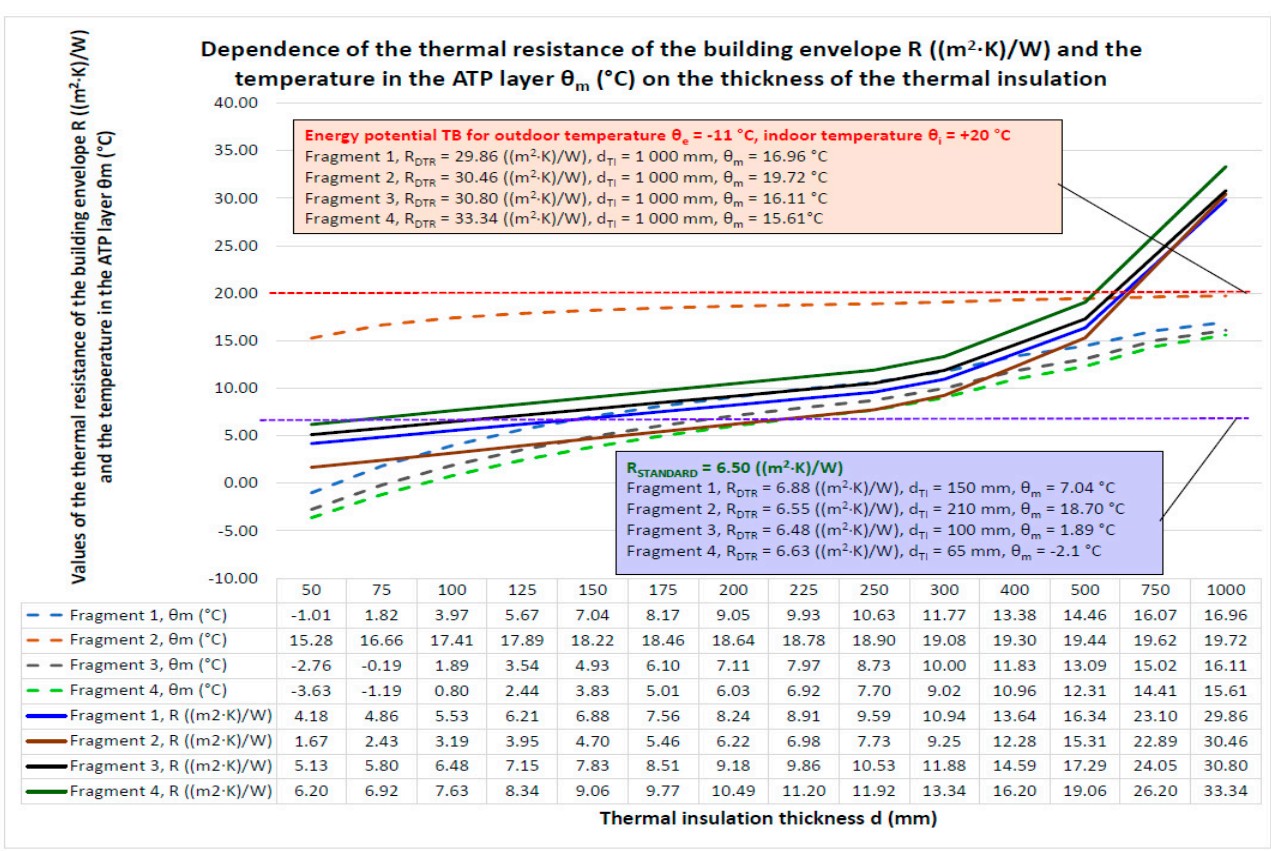

**Figure 39.** The graphical dependencies of the dynamic thermal resistance $R_{DTR}$ ((m²·K)/W) on the thickness of the static/dynamic thermal insulation and the mean temperature of the heat transfer medium in the ATP tubes $\theta_m$ (°C) for all fragments investigated.

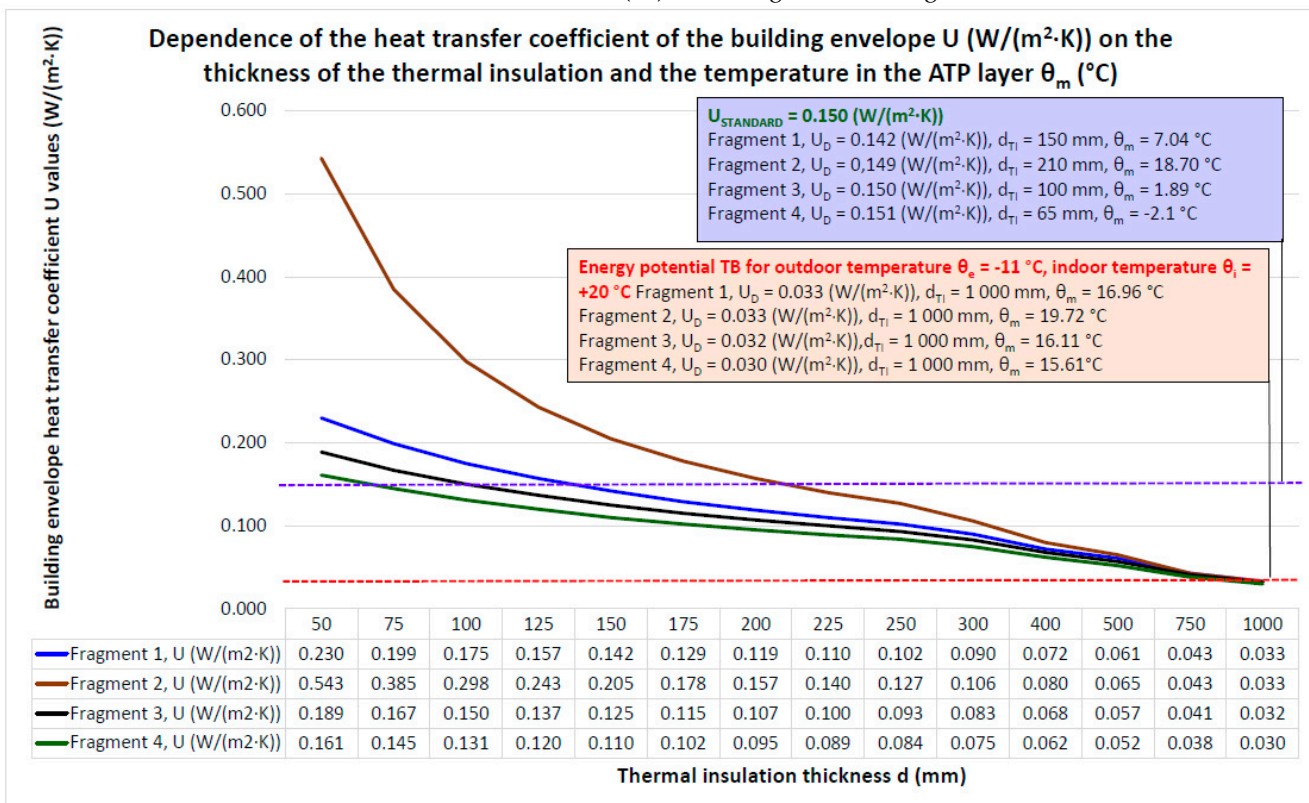

**Figure 40.** Graphical dependencies of the dynamic heat transfer coefficient $U$ (W/(m²·K)) on the thickness of the static/dynamic thermal insulation and the mean temperature of the heat transfer medium in the ATP tubes $\theta_m$ (°C) for all fragments investigated.

The contribution of our research lies in analyzing building structures with integrated energy-active elements and highlighting the application of active thermal protection in different energy functions based on the material composition of the building envelope. Important published results of our previous research in this area that form the basis for the analysis of the dynamic thermal barrier of building envelopes and the research carried out using the parametric study and computer simulation presented in this paper are included below.

In the article [4], 2021, the description of a novel technique and ATP application of thermal insulation panels with active heat transfer control in the form of a contact insulation system were the main points of emphasis. The innovative thermal insulation panels with integrated ATP (originator: Kalús) are a component of a lightweight prefabricated envelope that generates an indoor environment along with a low-temperature heating and high-temperature cooling system. RES or process waste heat is typically used as the energy source.

Among other contributions in this area is the evaluation of TB in terms of energy performance, economic efficiency, and ecology, [34], 2021. We compared the application of a classical envelope wall with standard thermal insulation meeting the requirements of the standards and an envelope wall with integrated TB with a significant elimination of the thickness of thermal insulation. We evaluated the application of the thermal barrier using three indicators: economic indicator one—a comparison between the money saved from thermal insulation at the conventional thickness and the cost of heat given to the TB in a structure with substantially less thermal insulation; economic indicator two—a comparison between the possible profit from the sale of the increased usable space of the building relative to the area at the standard thickness of thermal insulation and the price of heat delivered to the TB in a structure with drastically reduced thermal insulation; and economic indicator three—a comparison between the price of grey energy at the standard thickness of thermal insulation and the price of heat given to the TB in a structure with drastically reduced thermal insulation. The results showed that the thermal barrier represents a very efficient and promising solution in terms of the evaluation of economic indicators one to three, which are even more significant if we used heat for the TB from RES or waste heat.

In article [36], 2021, we described research on ITAP panels—interior thermally active panels with integrated active surface—that combine existing building and energy systems into one compact unit in an innovative way. These panels can be unified and manufactured as prefabricated products, which is their main advantage. This is associated with savings in production costs due to the technological manufacturing process, savings in installation costs during on-site implementation, and a reduction in implementation time due to the way they are used. They can be used in low-energy buildings that are suitable for the application of large-scale energy systems such as floor, wall, and ceiling large-scale heating/cooling. When comparing ITAP panels and walls with embedded large-area heating/cooling tubes on the inner wall, we came to the following conclusions: the thickness of the thermal insulation of ITAP panels (as required by the standard) has almost no effect in terms of energy demand when applied to the perimeter walls. In the case of an internal wall, which has a lower thermal resistance than the perimeter walls, the ITAP panels show savings of approximately 13% in heating and approximately 11% in cooling. For both systems, changing the pipe spacing from 100 to 150 mm reduces the performance by approximately 15 to 20%, and, for 200 mm spacing, the reduction in performance is approximately 30 to 35%. The effect of changing the pipe dimensions for both systems from 15 to 20 mm diameter is a variation of approximately 2.5%. The effect of the outside temperature or adjacent space temperature on the heating/cooling performance of both systems is a variation of approximately 5%, depending on the thermal insulation properties of the building structures on which these energy systems are applied.

In [35], 2022, we describe the design, project, and implementation of an experimental house, EB2020, in which we investigated the feasibility of using solar energy captured by

an energy solar roof (ESR) stored in ground-source heat storage (GSHS) to apply active thermal protection (ATP) in the building envelope for low-temperature radiant large-scale heating and thermal barrier (TB) function. In the theoretical calculations, the EB2020 house foundation slab as a GHS does not meet the requirements for low-temperature large-area radiant heating, but, as a heat source for TB in the building envelope, it is satisfactory. Experimental data made throughout one charging season and one discharging season were used to confirm the conclusions of the simplified calculations. We discovered that the building envelope's composition limits the amount of heat that can be used from the GHS in the experimental home after studying the results of theoretical calculations and experimental tests. TB significantly reduces heat loss/gain. Measurements have shown a highly energy-efficient use of ATP for passive cooling using TB with the use of the cold from long-term ground cold storage. With ambient temperatures reaching 34 °C, indoor air temperatures as low as 28 °C were recorded. Based on the findings of this research, only the shell of a building whose load-bearing component is constructed of a material with high thermal conductivity, i.e., low thermal resistance, such as reinforced concrete, can be advised to utilize GHS in conjunction with ATP and in the heating/cooling function.

In paper [31, 32], 2022, 2023, we outlined research on the development and improvement f building envelope panels that incorporate energy-active components that serve as a thermal barrier. Our research aimed to design and create a panel, drawing inspiration from the patented ®ISOMAX panel and system, whose design would be optimal in terms of thermal barrier operation and heat/cool accumulation. The patented system's manufacture of panels was overly complex and frequently displayed flaws in terms of design. We modeled both solar panels mathematically and physically and evaluated their energy potential. We created a unique panel using induction and analog molding. Most of the building components and all panels with integrated energy-active elements were produced directly in the prefabrication factory based on the synthesis of the knowledge derived from the scientific analysis and the transformation of these data. The prefabricated house IDA I was later realized as a prototype. The uniqueness of our ground-breaking building envelope panel system is in the panel design, which has 2.6 times less heat gain/loss than an ISOMAX panel.

In article [1], 2022, we presented a study of three variants of a reinforced concrete wall fragment insulated with thermal insulation from the exterior, with ATP pipes. The first variant has the ATP pipes on the interior side of the load-bearing layer of the structure, the second in the load-bearing reinforced concrete structure, and the third variant has the pipes located between the statically load-bearing and thermal insulation layer of the envelope structure. From the analysis, it is clear that for a pipe spacing of 50 to 100 mm in the thermal insulation, the heat flux to the interior $q_i$ (W/m²) increases, while the heat loss decreases rapidly. It was found that the thickness of the reinforced concrete core does not affect the heat flux as much as the thermal insulation. Based on the investigation, it can be concluded that the additional heat loss caused by Variant II, semi-accumulation heating (TABS system), and Variant III, accumulation heating, relative to Variant 1, direct heating, is minor, accounting for less than 1% of the total heat flux given. In the case of direct heating, the direct heat flux to the heated room is 89.17%, in the case of variant II (TABS) semi-accumulation heating is 73.36%, and in the case of variant III, it is 58.46%. Of the total heat flux delivered to the panel structure (TABS system), Variant II accounts for up to 14.84% and Variant III up to 29.86%.

Based on our research, not only articles but also three utility models have been developed (UM SK 5749 Y1 [37], UM SK 5729 Y1 [38], and UM SK 5725 Y1 [39]) and one European patent (EP 2 572 057 B1 [40]).

The novelty of this present work compared to the presented results of other authors and published results of our research lies in:

- A comprehensive analysis of the dynamic thermal barrier for the four most applied practices and materially different fragments of the building envelope;

- The creation of mathematical-physical models for the parametric study of the individual fragments under consideration;
- The creation of a 2D model for computer simulation of temperature distribution in individual layers of the building structure of a prefabricated timber building, the results of which will be verified on a test cell;
- The development of a graphical evaluation of the dynamic thermal resistance and dynamic heat transfer coefficient as a function of the temperature of the heat transfer medium in the TB layer for the comparison of the individual fragments;
- The evaluation of the multifunctional energy potential of individual building envelope fragments (TB functions, heating, cooling, and heat storage).

From a hypothetical point of view, just applying cold drinking water with an average temperature of $\theta_m$ = +10 °C to the ATP layer in Fragment 1 represents $R_{DTR}$ = 8.91 ((m²·K)/W), equivalent $U$ = 0.11 (W/(m²·K)), dynamic thermal insulation thickness 225 mm; in Fragment 3, it is $R_{DTR}$ = 11.883 ((m²·K)/W), equivalent $U$ = 0.083 (W/(m²·K)), dynamic thermal insulation thickness 300 mm; and in Fragment 4, it is $R_{DTR}$ = 14.773 ((m²·K)/W), equivalent $U$ = 0.0685 (W/(m²·K)), dynamic thermal insulation thickness 350 mm. The energy-saving potential of using TB is significant in both the heating and summer seasons. It is increased by the use of heat/cooling from RES and waste heat/cooling.

The basic input parameters entering the calculation of conventional and dynamic thermal resistance are described in the mathematical-physical model in Figure 41. In addition to the parameters characterizing the calculation of the conventional thermal resistance, the heat/cool $Q$ (kWh) delivered to the ATP at time $t$ (s) enters the calculation of the dynamic thermal resistance. This is influenced by the mean temperature of the heat transfer medium $\theta_m$ (°C) and the pipe spacing $L$ (m) in the ATP layer.

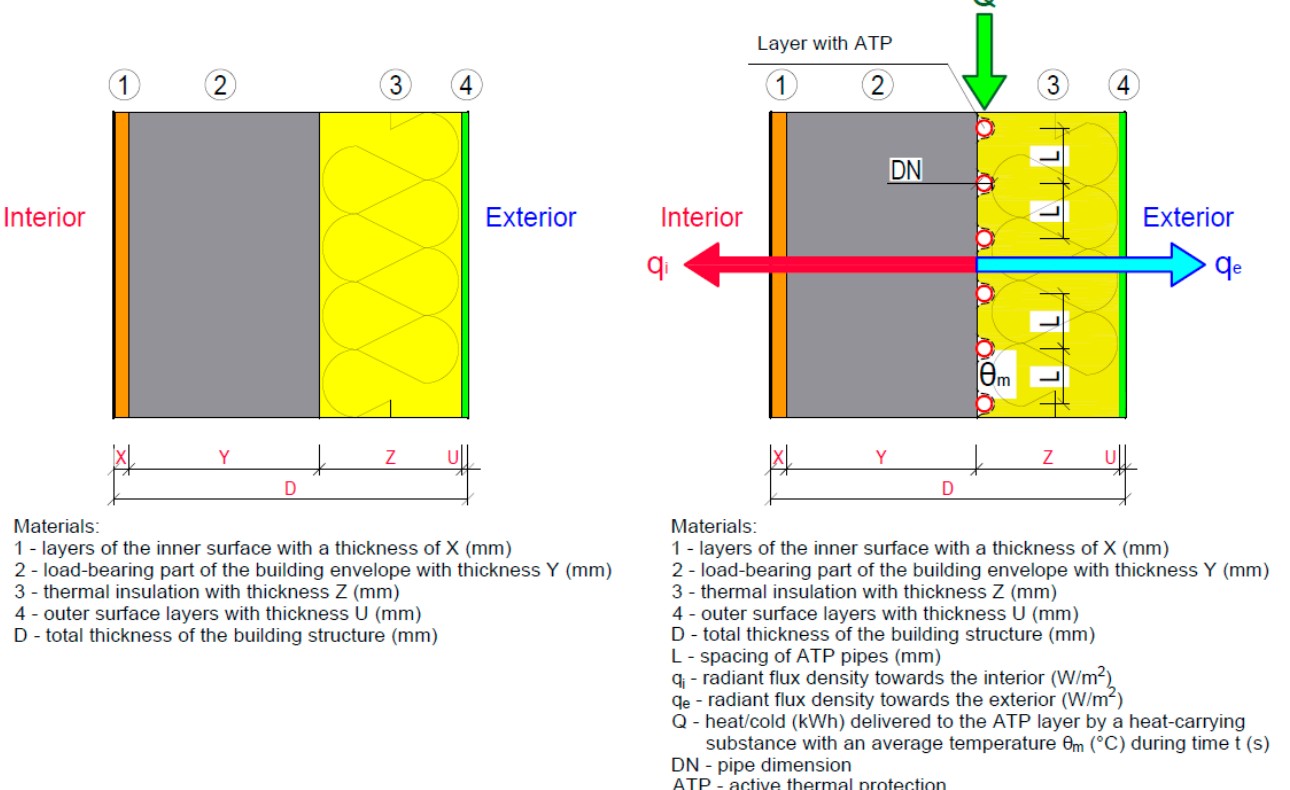

**Figure 41.** Mathematical-physical model for the calculation of conventional and dynamic thermal resistance.

Other significant research results in this area showing the high topicality and relevance of this issue have been published in the following articles and studies.

Mohammed Salah-Eldin Imbabi (2012) [41] simulated a wall with dynamic insulation based on exhaust air in winter and summer conditions. He arrived at the following results: in summer, the $U$-value of the dynamic insulation combined with natural and forced ventilation strategies was 73% ($U$ = 0.11 W/(m²·K)) and 65% ($U$ = 0.14 W/(m²·K)) lower than the static insulation $U$-value (0.4 W/(m²·K)), respectively. In winter, the average dynamic insulation U-value combined with forced and natural ventilation was 25% ($U$ = 0.3 W/(m²·K)) and 38% ($U$ = 0.25 W/(m²·K)) lower than the static thermal insulation $U$-value (0.4 W/(m²·K)).

Yaegashi, A., Hiyama, K., Kato, S., Tezuka, J., and Nikawa, S. [42] (2015) carried out measurements in an experimental wood building with applied DI. The actual U-value of the envelope was calculated to be 2.26 W/(m²·K) based on the measured heat loss in the case without DI. The heat loss per unit temperature on the surface with applied DI was 3.19 W/K without DI and 1.83 W/K with DI. The application of DI technology reduced heat loss by 42.6%.

Fantucci, S., Serra, V., and Perino, M. (2015) [43] investigated the performance of two configurations of dynamic airflow-based insulation systems in a climate chamber. The results showed that for the system with exhaust air in the heat recovery function, it is possible to achieve heat loss reductions ranging from 43 to 68%, and for the system with supply air in the preheat function, it is possible to achieve preheating efficiencies ranging from 9 to 20%, depending on the airflow velocity.

Vinay Shekar and Moncef Krarti [44], 2017, using a genetic algorithm-based optimization technique, identified the optimal R-value setting for commercial office buildings with dynamic insulation. The results of the analysis indicate that north-facing walls are more active and often need to change their R-value to minimize the energy consumption and cost of the office building. Savings optimally managing dynamic insulation could save up to 17% of the annual heating and cooling energy costs of U.S. office buildings.

Gopalan, A., Antony, A. S. M., Suresh, R., Sahoo, S., Livingston, L. M., Titus, A., and Sathyamurthy, R. [45], 2022, investigated the combined effect of two technologies: a phase change material and a dynamic insulation system. They found that a wall with both technologies integrated can provide a 15 to 72% reduction in annual radiant heat, then a 7 to 38% reduction in heat loss depending on the environment.

ATP in buildings can also be used in the cooling function. It can also use waste heat. Buildings need to be comprehensively assessed for their sustainability impact. Important scientific works in these areas include publications [46–48].

Park, B., Srubar, W., V., and Krarti, M. [49] (2015) conducted a comparative analysis using a simulation environment capable of modeling DIM to evaluate the impact of DIM on the final energy consumption for heating and cooling single-zone residential buildings in three U.S. climate regions. The results showed that variable thermal resistance envelope materials (RSI-0.5/RSI-2.5) can reduce annual cooling energy consumption in residential buildings by an average of 15% and up to 39% in all 3 U.S. climates and annual heating energy consumption by an average of 10% in moderate U.S. climates, depending on window size and internal heat gains.

Menyhart, K. and Krarti, M. [50] (2017) presented results from a comprehensive analysis of potential energy savings for heating and cooling due to the replacement of conventional static insulation with dynamic insulation materials for residential buildings in the U.S. The authors used two control schemes to trigger the switching mechanism: using temperature and a measured temperature profile based on climate and weather trends. The largest savings were achieved with climate-based control, with overall energy savings ranging from 7 to 42%.

Pflug, T., Bueno, B., E., Siroux, M., and Kuhn T., E. [51] (2017) investigated the possibilities of a façade element with switchable U-value and g-value. A first prototype was constructed and measured in an isolated condition, and the properties of the façade

element were optimized using theoretical analysis at the façade level. The U-value can be changed from a value of 0.35 W m$^{-2}$ K$^{-1}$ in the insulating state to a value of 2.7 W m$^{-2}$ K$^{-1}$ in the conductive state. Building energy simulations using EnergyPlus showed that the sum of heating and cooling requirements could be reduced by approximately 30% if the switchable insulation was used as a window application or in front of an opaque solid wall.

Garriga Martínez, M.S., Dabbagh M., and Krarti, M. [52] (2019) evaluated the potential energy cost savings by applying static and dynamic insulation materials (DIM) for three prototype dwellings in Spain. The results of the analysis show that DIMs with the largest R-value step (i.e., the difference between high and low R-values) achieve the highest source energy savings, reaching up to 19% source energy reduction for heating and cooling for the entire housing stock in Barcelona. At the same time, the reduction in peak electricity consumption associated with the retrofit of the external walls for the existing housing stock may lead to no need for new power plant construction. The use of DIM for the existing housing stock in Barcelona could reduce annual $CO_2$ emissions by more than 300,000 tonnes or 6.80% of the total amount of $CO_2$ currently emitted to heat and cool houses.

Rupp, S. and Krarti, M. [53] (2019) built on past DIM studies that investigated binary control of the wall R-value (R system on or off). They added a period during which the wall R-value can vary continuously within a defined range. The optimized mode of operation for a residential building in the state of Colorado (U.S.A.) demonstrated heating energy savings of 9.3% and cooling energy savings of 21.0% compared to RSI-3.8 static insulation. The authors also performed a parametric analysis for several climatic regions in the U.S.A. They concluded that, especially in areas with few cooling days, there is the potential to save up to 35% cooling energy, while in climates with few heating days, the potential for significant heating energy reductions of more than 80% compared to static insulation was found.

In Kishore, R. A, Bianchi, M., V., A., Booten, Ch., Vidal, J, and Jackson, R.'s study [54] (2021), the subject of the research was a novel wall structure composed of a layer of phase change material (PCM) between two layers of dynamic insulating material and system (DIMS). The authors concluded that a wall integrated with PCM-DIMS provides greater energy-saving opportunities than either a wall integrated with DIMS alone or a wall integrated with PCM alone in all climates and wall orientations analyzed in this study. Depending on the climate, a wall integrated with PCM-DIMS could provide a 15–72% reduction in annual heat gains and a 7–38% reduction in annual heat losses.

## 6. Conclusions

In this section, we summarize the most important outcomes of our research:

- We developed mathematical-physical models for four materially different building envelope types to determine the energy saving and energy storage potential of ATP, as well as to define the dynamic thermal resistance using a parametric study;
- Due to the application of thermal insulation also on the interior side in Fragment 1, the function of the ATP for this building envelope solution is limited to the thermal barrier and heat/cool accumulation functions only;
- Because the load-bearing wall is made of porous concrete blocks, Fragment 3 has a high thermal resistance, and the function of the ATP for this building envelope solution is limited only to the functions of a thermal barrier and partial heat/cold accumulation;
- Based on the analysis of the dynamic thermal resistance, we can conclude that in the case of the building envelope, Fragment 4, ATP is significant only as a function of TB, but at relatively low mean temperatures of the heat carrier $\theta_m$ = 15.61 °C, the dynamic thermal resistance has a high-value $R_{DTR}$ = 30.34 ((m$^2$·K)/W), which corresponds to a static thermal insulation thickness of 1000 mm;

- An important result of the computer simulation is the uniform and continuous temperature distribution in the ATP heat transfer layer with a temperature $\theta_m$ = of about 6 °C, confirming the functionality of the TB with the achievement of a dynamic thermal resistance $R_{DTR}$ = 10.487 ((m²·K)/W) at a dynamic thermal insulation thickness of 100 mm, which is equal to the thermal resistance at a static thermal insulation thickness of 200 mm;
- The relatively low mean temperature of the heat transfer medium $\theta_m$ = 15.61 to 19.72 °C delivered to the tubes of the ATP heat transfer layer gives a dynamic thermal resistance of $R_{DTR}$ = 29.86 to 33.34 ((m²·K)/W) with an equivalent dynamic thermal insulation thickness of 1000 mm for the required standard resistances $R_{STANDARD}$ = 6.50 ((m²·K)/W) of the individual fragments of the building envelope with static thermal insulation of 65 to 210 mm. Then, the energy potential of using TB is 455 to 513% for the increase in thermal resistance and 476 to 1.538% for the thickness of the dynamic thermal insulation;
- A mean temperature of the heat transfer medium $\theta_m$ (°C) delivered to the tubes of the ATP heat transfer layer equal to the interior temperature $\theta_i$ (°C) represents zero heat loss/gain to and from the interior;
- The energy-saving potential of using TB is undoubtedly significant in the heating season as well as in the summer season. It is increased by the use of heating/cooling from RES;
- The computer simulation was intended only for a basic analysis of the uniform and continuous temperature distribution in the ATP layer. We will continue the simulations to analyze for different changes in input parameters, the heat fluxes to the interior and exterior, the amount of heat delivered, the effect of operating time, and other physical variables affecting the dynamic thermal resistance of the individual building envelope structures.

In the near future, in the field of active thermal protection, we are preparing research on energy-multifunctional building envelopes using complex computer simulation and experimental verification on a test cell. We are preparing prototypes of panels that will have the function of heating/cooling, solar energy absorption, and ambient energy absorption in addition to the function of the thermal barrier. We are also preparing a prototype of an insulation panel with an integrated PV area for electricity generation, with an integrated thermal barrier and a DHW preheating register serving simultaneously in summer to eliminate heat gains and cool the PV area.

## 7. Patents

Based on our research and work reported in this manuscript, there resulting are three utility models have been developed (UM SK 5749 Y1 [37], UM SK 5729 Y1 [38], and UM SK 5725 Y1 [39]) and one European patent (EP 2 572 057 B1 [40]).

**Author Contributions:** Conceptualization, V.M. and D.K. (Daniel Kalús); methodology, V.M. and D.K. (Daniel Kalús); software—excel, V.M. and D.K. (Daniel Kalús); validation, V.M., D.K. (Daniel Kalús), D.K. (Daniela Koudelková), M.K., Z.S., R.I., M.S., and P.Š.; formal analysis, V.M., D.K. (Daniel Kalús), D.Ko. (Daniela Koudelková), M.K., Z.S., and R.I.; investigation, V.M., D.K. (Daniel Kalús), D.K. (Daniela Koudelková), M.K., Z.S., and R.I.; resources, V.M., D.K. (Daniel Kalús), D.K. (Daniela Koudelková), M.K., Z.S., R.I., M.S., and P.Š.; data curation, V.M. and D.K. (Daniel Kalús); writing—original draft preparation, V.M., D.K. (Daniel Kalús), D.K. (Daniela Koudelková), M.S., and P.Š.; writing—review and editing, V.M., D.K. (Daniel Kalús), D.K. (Daniela Koudelková), M.S., and P.Š.; visualization, V.M. and M.S.; supervision, V.M. and D.K. (Daniel Kalús); project administration, V.M., M.K., M.S., and P.Š.; funding acquisition, D.K. (Daniel Kalús) and R.I. All authors have read and agreed to the published version of the manuscript.

**Funding:** The research was supported by the private company EHBconsulting, s.r.o.

**Institutional Review Board Statement:** Not applicable.

**Informed Consent Statement:** Not applicable.

**Data Availability Statement:** Not applicable.

**Acknowledgments:** This research is supported by the Ministry of Education, Science, Research and Sport of the Slovak Republic through the grant VEGA 1/0229/21: Building physics fundamentals of a nearly zero energy building related to its environmental aspects.

**Conflicts of Interest:** The authors declare no conflict of interest.

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
