# Peer review of "Analysis of the Dynamic Thermal Barrier in Building Envelopes"

_coatings, doi:10.3390/coatings13030648_

Round 1
Reviewer 1 Report
Abstract need to change
Introduction part less number of the paper was identified. Kindly add three or four latest paper
It is observed many spelling and grammer mistake is there
In materials and methods section is not clarity
Results and discussion more supporting reference was required
conclusion need to be change
Author Response
We accept the reviewer's comments and suggestions in their entirety and thank you for your valuable advice and guidance to improve our article.
We have attempted to restyle the present manuscript and make major revisions according to the comments of all the reviewers so that it is clearer and clearly declares the various information related to the present research in isolation but also coherently in relation to each other. We have also added the scientific explanation/mechanism - novelty and contributions of our research described in this paper. All changes made to the article are indicated in red font (Manuscript with changes indicated). We provide the following responses to each comment:
Point 1: Abstract need to change.
Response 1: We have reviewed the abstract and made the necessary corrections and edits.
Point 2: Introduction part less number of the paper was identified. Kindly add three or four latest paper.
Response 2: A review of the scientific and professional literature in this area is presented in Chapter 2. For this reason, we have not added references to other papers in the introductory section.
Point 3: It is observed many spelling and grammer mistake is there.
Response 3: We have reviewed the article and made the necessary language corrections and modifications.
Point 4: In materials and methods section is not clarity.
Response 4: In Chapter 3, we have added a more autumnal formulation of the research objective. The research described in this paper aims to analyze the dynamic thermal barrier in building envelopes, which is characterized by the dynamic thermal resistance depending on the temperature change of the heat transfer medium supplied to the dynamic TB layer and to determine the energy potential of several materially different fragments of the building envelope. The TB and DTR functions depend on uniform and continuous temperature maintenance in a given layer of the building structure. The research methodology is based on the analysis and synthesis of procedures for calculating thermal resistance and heat transfer coefficient according to EN 73 0540, wall heating according to EN 1264-(1-5) and computer simulation in ANSYS.
Point 5: Results and discussion more supporting reference was required.
Response 5: In Chapter 5, based on the request of one of the reviewers, we have added other significant research results in this area, which show the high topicality and importance of this issue.
Point 6: Conclusion need to be change.
Response 6: We have reviewed the conclusion, simplified it and made the necessary corrections and adjustments. Based on a request from one of the reviewers, we are adding information on upcoming research in the area of active thermal protection. In the near future, in the field of active thermal protection, we are preparing research on energy-multifunctional building envelopes by means of complex computer simulation and experimental verification on a test cell. We are preparing prototypes of panels that will have the function of heating/cooling, solar energy absorber and ambient energy absorber in addition to the function of the thermal barrier. We are also preparing a prototype of an insulation panel with an integrated PV area for electricity generation, with an integrated thermal barrier and a DHW preheating register serving simultaneously in summer to eliminate heat gains and cool the PV area.

Reviewer 2 Report
This work analyzed that a parametric study of the dynamic thermal resistance, simulation of the progression of the temperature in the individual layers of the structure and determination of the energy potential of several materially different fragments of building envelope wall structures with integrated energy-active elements under different input data of physical variables. In some cases, the text must be clarified and there are issues that are not accurate.
1. Image resolution of Fig.1~3 can be improved.
2. The section 2 is too tedious, please simplified it and it’s better to dived several parts, e.g., 2.1 XXXX; 2.2XXXX; 2.3 XXXX.
3. The accuracy of the computer simulation can be illustrated.
4. The fonts in each figure should be uniform.
5. Conclusions can be simplified.
6. The authors may give an outlook in the field of active thermal protection.
Author Response
We accept the reviewer's comments and suggestions in their entirety and thank you for your valuable advice and guidance to improve our article.
We have attempted to restyle the present manuscript and make major revisions according to the comments of all the reviewers so that it is clearer and clearly declares the various information related to the present research in isolation but also coherently in relation to each other. We have also added the scientific explanation/mechanism - novelty and contributions of our research described in this paper. All changes made to the article are indicated in red font (Manuscript with changes indicated). We provide the following responses to each comment:
Point 1: Image resolution of Fig.1~3 can be improved.
Response 1: We have tried to improve the resolution of the images. However, some images are taken with a link to a specific web page where it was no longer possible to improve their sharpness and quality.
Point 2: The section 2 is too tedious, please simplified it and it’s better to dived several parts, e.g., 2.1 XXXX; 2.2XXXX; 2.3 XXXX.
Response 2: We have made the necessary corrections, and modifications and shortened Chapter 2. We have also divided it into 2.1 Inspirational technical solutions of our research, 2.2 Scientific and professional work in the field of ATP, and 2.3 Scientific works in the field of ATP computer simulations based on a comment by one of the reviewers.
Point 3: The accuracy of the computer simulation can be illustrated.
Response 3: Only a simple computer simulation has been made to show the temperature evolution in the structure. The calculation of the temperatures in the structure with known mathematical and physical relations for the calculation of the temperatures in the structure served as a control. In the paper, the waveform of temperatures calculated from the relations and also from the computer simulation is described and indicated in the figures.
Point 4: The fonts in each figure should be uniform.
Response 4: Due to the fact that the images are from different sources or of different sizes, it is not possible to achieve absolute uniformity in fonts. We have tried to modify as many images as possible so that the font is uniform.
Point 5: Conclusions can be simplified.
Point 6: The authors may give an outlook in the field of active thermal protection.
Response 5 and 6: We have combined the responses to comments 5 and 6 into a single response as they are closely related.
We have reviewed the conclusion, simplified it and made the necessary corrections and adjustments. Based on a request from one of the reviewers, we are adding information on upcoming research in the area of active thermal protection. In the near future, in the field of active thermal protection, we are preparing research on energy-multifunctional building envelopes by means of complex computer simulation and experimental verification on a test cell. We are preparing prototypes of panels that will have the function of heating/cooling, solar energy absorber and ambient energy absorber in addition to the function of the thermal barrier. We are also preparing a prototype of an insulation panel with an integrated PV area for electricity generation, with an integrated thermal barrier and a DHW preheating register serving simultaneously in summer to eliminate heat gains and cool the PV area.
In Chapter 5, based on the request of one of the reviewers, we have added other significant research results in this area, which show the high topicality and importance of this issue.

Reviewer 3 Report
The current paper reports the analysis of thermal protection of a thermal barrier in a building envelope structure. The writing style is poor and clunky. So many information are presented but not in a coherent manner. From a non-review paper point of view, this present paper is too cluttered and full of information, that does not necessarily link to the present investigation. In addition, there is hardly any scientific explanation/mechanism beyond the presented experimental data. Based on that, I recommend major revision of the manuscript. The specific comments are as follows:
1. The paper require a moderate revision from grammar and language point of view.
2. The title can be revised and concise. In current form it is bit bulky!
3. There are too many paragraphs in section 1 with few lines in each of them. Better to combine the similar information in a given paragraph and make it smooth for the reader.
4. Section 2 is way too long for the non-review paper. Please shorten it and even may consider to combine it with section 1.
5. Instead of mentioning the contribution of each individual author groups (section 2, mainly) separately, try to analyse the results in a critical way to find the gaps/further contribution in this field.
6. What’s the purpose of section 4.3? It is just the summary of previously reported works. From my point of view, it is not adding any value of your present paper. If you must, then at beast you can co-relate the findings of the present paper with the earlier reported work and present the novelty of the present work.
7. The conclusion section is also way too long. Try to concise in view of other published papers.
8. There are so many unnecessary information are present that are not linked with the focus of the paper. Only retain the information that are directed related to the present investigation.
Author Response
We accept the reviewer's comments and suggestions in their entirety and thank you for your valuable advice and guidance to improve our article.
We have attempted to restyle the present manuscript and make major revisions according to the comments of all the reviewers so that it is clearer and clearly declares the various information related to the present research in isolation but also coherently in relation to each other. We have also added the scientific explanation/mechanism - novelty and contributions of our research described in this paper. All changes made to the article are indicated in red font (Manuscript with changes indicated). We provide the following responses to each comment:
Point 1: The paper require a moderate revision from grammar and language point of view.
Response 1: We have reviewed the article and made the necessary language corrections and modifications.
Point 2: The title can be revised and concise. In current form it is bit bulky!
Response 2: We have revised and concise the title of the article.
Point 3: There are too many paragraphs in section 1 with few lines in each of them. Better to combine the similar information in a given paragraph and make it smooth for the reader.
Response 3: We have made the necessary corrections and modifications to section 1 and have minimised paragraphs for clarity.
Point 4: Section 2 is way too long for the non-review paper. Please shorten it and even may consider to combine it with section 1.
Response 4: We have made the necessary corrections, and modifications and shortened Chapter 2. We have also divided it into 2.1 Inspirational technical solutions of our research, 2.2 Scientific and professional work in the field of ATP, and 2.3 Scientific works in the field of ATP computer simulations based on a comment by one of the reviewers.
Point 5: Instead of mentioning the contribution of each individual author groups (section 2, mainly) separately, try to analyse the results in a critical way to find the gaps/further contribution in this field.
Point 6: What’s the purpose of section 4.3? It is just the summary of previously reported works. From my point of view, it is not adding any value of your present paper. If you must, then at beast you can co-relate the findings of the present paper with the earlier reported work and present the novelty of the present work.
Response 5 and 6: We have combined the responses to comments 5 and 6 into a single response as they are closely related.
Chapter 2 provides a broader description of research in this area. In Chapter 5, we present important research outputs from these aforementioned papers (from Chapter 2) and describe the published results of our previous research in this area (previously presented in Section 4.3), which form the basis for the analysis of the dynamic thermal barrier of building envelopes and the research carried out using the parametric study and computer simulation presented in this paper. The contribution of our research lies in the analysis of building structures with integrated energy-active elements and in highlighting the application of active thermal protection in different energy functions based on the material composition of the building envelope.
The novelty of the present work compared to the presented results of other authors and published results of our research lies in:
- a comprehensive analysis of the dynamic thermal barrier for the 4 most practically applied and materially different fragments of the building envelope,
- the creation of mathematical-physical models for the parametric study of the individual fragments under consideration,
- creation of a 2D model for computer simulation of temperature distribution in individual layers of the building structure of a prefabricated timber building, the results of which will be verified on a test cell,
- the development of a graphical evaluation of the dynamic thermal resistance and dynamic heat transfer coefficient as a function of the temperature of the heat transfer medium in the TB layer for the comparison of the individual fragments; and
- the evaluation of the multifunctional energy potential of individual building envelope fragments (TB functions, heating, cooling, heat storage).
Point 7: The conclusion section is also way too long. Try to concise in view of other published papers.
Point 8: There are so many unnecessary information are present that are not linked with the focus of the paper. Only retain the information that are directed related to the present investigation.
Response 7 and 8: We have combined the responses to comments 7 and 8 into a single response as they are closely related.
We have reviewed the conclusion, simplified it and made the necessary corrections and adjustments. Based on a request from one of the reviewers, we are adding information on upcoming research in the area of active thermal protection. In the near future, in the field of active thermal protection, we are preparing research on energy-multifunctional building envelopes by means of complex computer simulation and experimental verification on a test cell. We are preparing prototypes of panels that will have the function of heating/cooling, solar energy absorber and ambient energy absorber in addition to the function of the thermal barrier. We are also preparing a prototype of an insulation panel with an integrated PV area for electricity generation, with an integrated thermal barrier and a DHW preheating register serving simultaneously in summer to eliminate heat gains and cool the PV area.
In Chapter 5, based on the request of one of the reviewers, we have added other significant research results in this area, which show the high topicality and importance of this issue.

Round 2
Reviewer 1 Report
Accept in Present Form
Reviewer 2 Report
Accept in present form
Reviewer 3 Report
Can be accpeted in current form